# Circuits and mechanisms for TMS-induced corticospinal waves: Connecting sensitivity analysis to the network graph

Gene J. Yu[1,2]*, Federico Ranieri[3], Vincenzo Di Lazzaro[4,5], Marc A. Sommer[1], Angel V. Peterchev[1,2,6,7], Warren M. Grill[1,6,7,8]

**1** Department of Biomedical Engineering, Duke University, Durham, North Carolina, United States of America, **2** Department of Psychiatry and Behavioral Sciences, Duke University, Durham, North Carolina, United States of America, **3** Neurology Unit, Department of Neuroscience, Biomedicine and Movement Sciences, University of Verona, Verona, Italy, **4** Department of Medicine and Surgery, Unit of Neurology, Neurophysiology, Neurobiology and Psychiatry, Università Campus Bio-Medico di Roma, Roma, Italy, **5** Fondazione Policlinico Universitario Campus Bio-Medico, Roma, Italy, **6** Department of Electrical and Computer Engineering, Duke University, Durham, North Carolina, United States of America, **7** Department of Neurosurgery, Duke University, Durham, North Carolina, United States of America, **8** Department of Neurobiology, Duke University, Durham, North Carolina, United States of America

* gene.yu@duke.edu

**Data Availability Statement:** The source code and data used to produce the results and analyses presented in this manuscript are available in a Github repository at https://github.com/

## Abstract

Transcranial magnetic stimulation (TMS) is a non-invasive, FDA-cleared treatment for neuro-psychiatric disorders with broad potential for new applications, but the neural circuits that are engaged during TMS are still poorly understood. Recordings of neural activity from the corticospinal tract provide a direct readout of the response of motor cortex to TMS, and therefore a new opportunity to model neural circuit dynamics. The study goal was to use epidural recordings from the cervical spine of human subjects to develop a computational model of a motor cortical macrocolumn through which the mechanisms underlying the response to TMS, including direct and indirect waves, could be investigated. An in-depth sensitivity analysis was conducted to identify important pathways, and machine learning was used to identify common circuit features among these pathways. Sensitivity analysis identified neuron types that preferentially contributed to single corticospinal waves. Single wave preference could be predicted using the average connection probability of all possible paths between the activated neuron type and L5 pyramidal tract neurons (PTNs). For these activations, the total conduction delay of the shortest path to L5 PTNs determined the latency of the corticospinal wave. Finally, there were multiple neuron type activations that could preferentially modulate a particular corticospinal wave. The results support the hypothesis that different pathways of circuit activation contribute to different corticospinal waves with participation of both excitatory and inhibitory neurons. Moreover, activation of both afferents to the motor cortex as well as specific neuron types within the motor cortex initiated different I-waves, and the results were interpreted to propose the cortical origins of afferents that may give rise to certain I-waves. The methodology provides a workflow for performing computationally tractable sensitivity analyses on complex models and relating the results to the network structure to both identify and understand mechanisms underlying the response to acute stimulation.

genejongyu/tms-iwaves-structuralanalysis-2024. The full data generated during optimization and analysis are on Zenodo at https://doi.org/10.5281/zenodo.10729433.

**Funding:** The research reported in this publication was supported by the National Institute of Neurological Disorders and Stroke of the National Institutes of Health under Award Number R01NS117405 (GJY, MAS, AVP, WMG). The content is solely the responsibility of the authors and does not necessarily represent the official views of the National Institutes of Health. The funders had no role in study design, data collection and analysis, decision to publish, or preparation of the manuscript.

**Competing interests:** A.V.P. is an inventor on patents on TMS technology and has received equity options, scientific advisory board membership, and consulting fees from Ampa Health; patent royalties and consulting fees from Rogue Research; consulting fees and equity options from Magnetic Tides; and consulting fees from Soterix Medical; equipment loan from MagVenture; and research funding from Motif.

## Author summary

Understanding circuit mechanisms underlying the response to transcranial magnetic stimulation remains a significant challenge for translational and clinical research. Computational models can reconstruct network activity in response to stimulation, but basic sensitivity analyses are insufficient to identify the fundamental circuit properties that underly an evoked response. We developed a data-driven neuronal network model of motor cortex, constrained with human recordings, that reproduced the corticospinal response to magnetic stimulation. The model supported several hypotheses, e.g., the importance of stimulating incoming fibers as well as neurons within the cortical column and the relevance of both excitatory and inhibitory neurons. Following a sensitivity analysis, we conducted a secondary structural analysis that linked the results of the sensitivity analysis to the network using machine learning. The structural analysis pointed to anatomical mechanisms that contributed to specific peaks in the response. Generally, given the anatomy and circuit of a neural region, identifying strongly connected paths in the network and the conduction delays of these paths can screen for important contributors to response peaks. This work supports and expands on hypotheses explaining the response to transcranial magnetic stimulation and adds a novel method for identifying generalizable neural circuit mechanisms.

## Introduction

Transcranial magnetic stimulation (TMS) can non-invasively activate superficial cortical regions to study brain functions, treat psychiatric and neurological disorders, and collect diagnostic biomarkers [1]. However, improving methodologies and developing new applications remain slow and challenging due to the uncertainties about what is activated by TMS and how this activation courses through the circuits within and beyond the stimulated region [2]. One approach to understanding these network effects in the motor cortex is via descending volleys of activity that propagate to the spinal cord in response to TMS and can be recorded epidurally as transient corticospinal waves (Fig 1). The corticospinal waves represent the activity of layer 5b pyramidal tract neurons (PTNs) that send axons into the spinal cord [3]. The shortest latency direct wave (D-wave) is widely agreed to represent the direct activation of PTNs [4]. Subsequent waves are called indirect waves (I-waves) and likely represent transsynaptic activations of PTNs resulting from the initial direct activation of PTNS, axons of afferents, and other neuron types. Understanding the neurons and circuits that produce the I-waves would provide insight into patterns of neuron activation and the circuit connections that mediate the cortical response to TMS [5].

Current understanding of I-waves arises from epidural recordings combined with pharmacological interventions that identified the synaptic receptors involved in I-wave generation and broadly suggested excitatory and inhibitory mechanisms that contribute to I-waves [5,6]. These and other experimental findings were organized into conceptual frameworks to propose mechanisms that give rise to the corticospinal waves [5,6]. Two broad categories of these frameworks are I-wave generation through circuit activations and I-wave generation via intrinsic neuronal mechanisms (neural oscillator hypothesis). With circuit activation, cortico-cortical afferents are thought to initiate activations in different neuronal populations that propagate through the cortical circuit to L5 PTNs. Intrinsic neuronal mechanisms have also been hypothesized to allow L5 PTNs to behave as neural oscillators such that the I-waves result

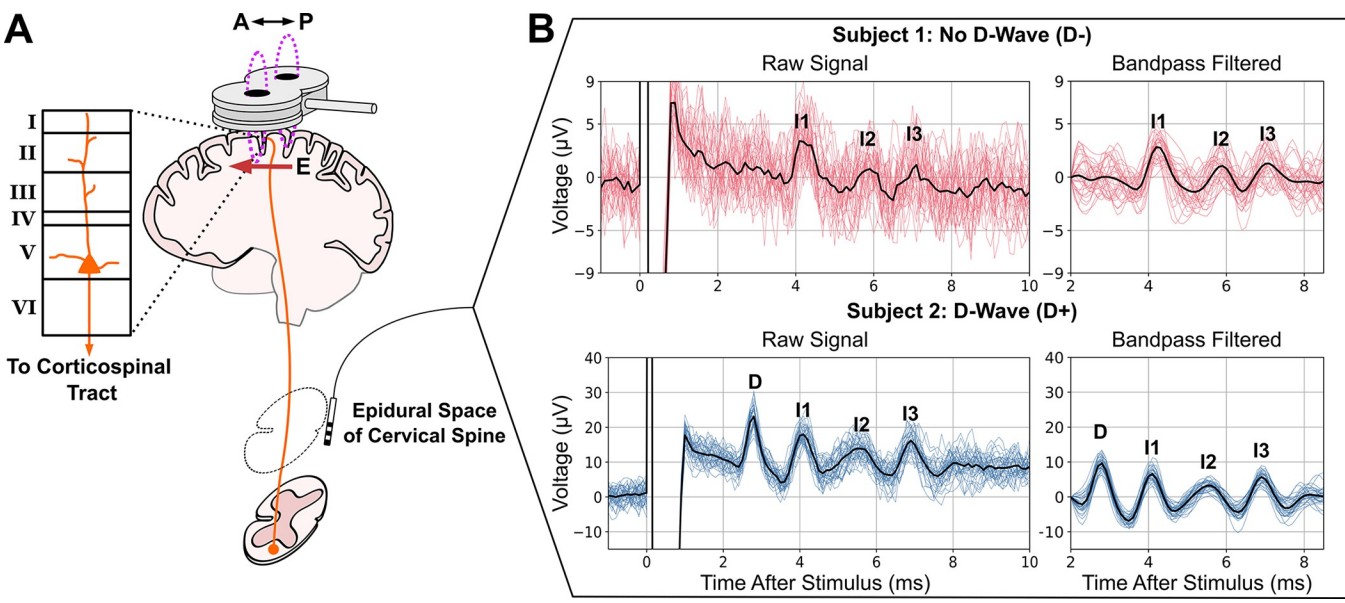

**Fig 1. Descending volleys of spinal waves provide a window into motor cortical responses to TMS.** A) TMS coil with electric field (E) induced in the posterior–anterior (P–A) orientation over the motor cortex. L5 PTNs send axons into the spinal cord (corticospinal tract), and their activity is recorded epidurally at levels C1–C5. B) Epidural recordings of corticospinal waves in two human subjects. Individual trials are plotted with colored lines. The solid black lines are trial averages.

from repeated spiking from the same neuron due to the dynamics following initial excitation by TMS.

Computational neuronal network models have been developed that integrate anatomical and electrophysiological details to investigate TMS-induced corticospinal waves. A model by Esser et al. represented the major layers of motor cortex using spiking point neurons and homogeneous activation of a proportion of fiber terminals across all layers to represent activation by single TMS pulses [7]. Rusu et al. developed a network model of layer 2/3 and layer 5 pyramidal neurons with realistic dendritic morphologies to investigate the effect of somato-dendritic conduction and integration on I-wave generation [8]. These models generated I-wave activity that qualitatively resembled experimental findings. However, the models were not directly constrained by experimental recordings and lacked an exhaustive sensitivity analysis to investigate, among other variables, the effects of inhomogeneous activation across different neuron types.

To determine the TMS activations and neuron-to-neuron projections that contribute to I-waves, we used experimental recordings of the corticospinal response to TMS to provide objectives to optimize a computational model of a motor cortical macrocolumn. Starting from a reduced version of the Esser model, that could produce I-waves and is mathematically compact, we established a spiking neuronal network model of motor cortex that reproduced the features of D-waves and I-waves recorded epidurally in the cervical spine of human subjects. Next, a unified model was developed that generated responses with and without a D-wave with a change in a single parameter. A sensitivity analysis of the unified model was conducted using the two-variable-at-a-time (TVAT) method. Finally, machine learning and graph theoretical measures were used to relate the connectivity of the model to the results of TVAT analysis and identify general mechanisms producing I-waves at the circuit level. A high-level representation of the methodology is summarized in Fig 2.

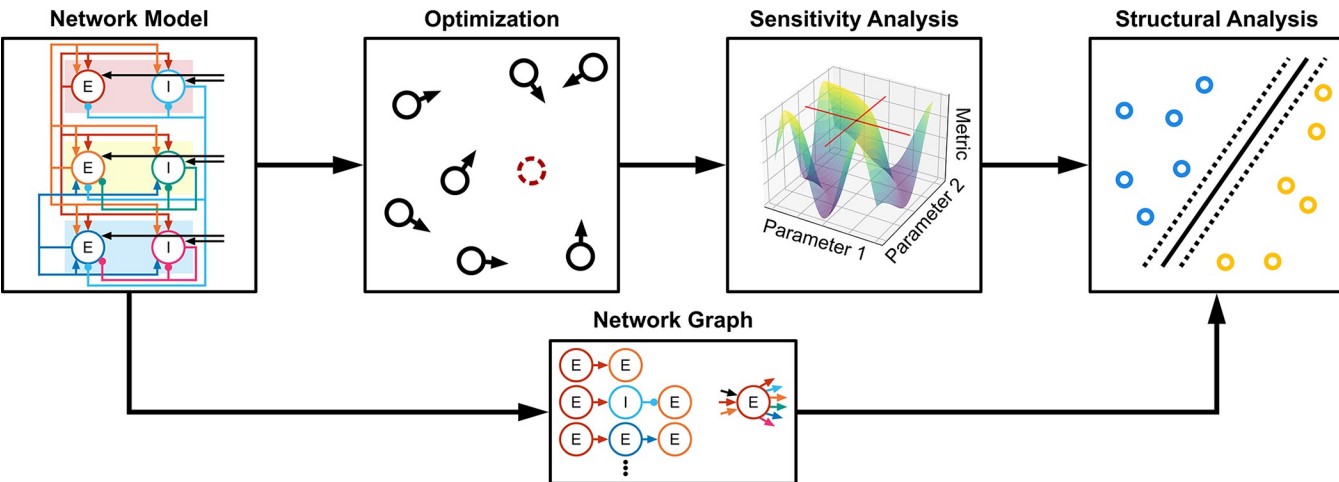

**Fig 2. High level diagram of methodology.** A network model was defined, and particle swarm optimization was used to constrain parameters using experimental data. A TVAT sensitivity analysis was conducted on the optimized model, and finally the network graph was used to identify structural patterns that predict the sensitivity analysis. E: Excitatory neuron. I: Inhibitory neuron.

# Results

The neuronal network model used to simulate the effects of TMS represents a human cortical macrocolumn within the motor cortex and included layer (L) 2/3, L5 and L6 and is based on a model developed in Esser et al., 2005 [7] (Fig 3A). Each layer contained excitatory neurons representing pyramidal neurons and inhibitory neurons representing fast-spiking parvalbumin-positive basket cells (BC). More specifically, the layer 2/3 and layer 6 pyramidal neurons were intratelencephalic (IT) neurons, while the layer 5 pyramidal neurons were PTNs. Inhibition was mediated only by parvalbumin-positive BCs because they provide the strongest inhibition compared to somatostatin and vasoactive intestinal protein expressing interneurons [9]. Excitatory afferents (AFF) were included that targeted each of the neuron types. The afferents non-specifically represented activity that arise from other cortical/sub-cortical areas. Direct

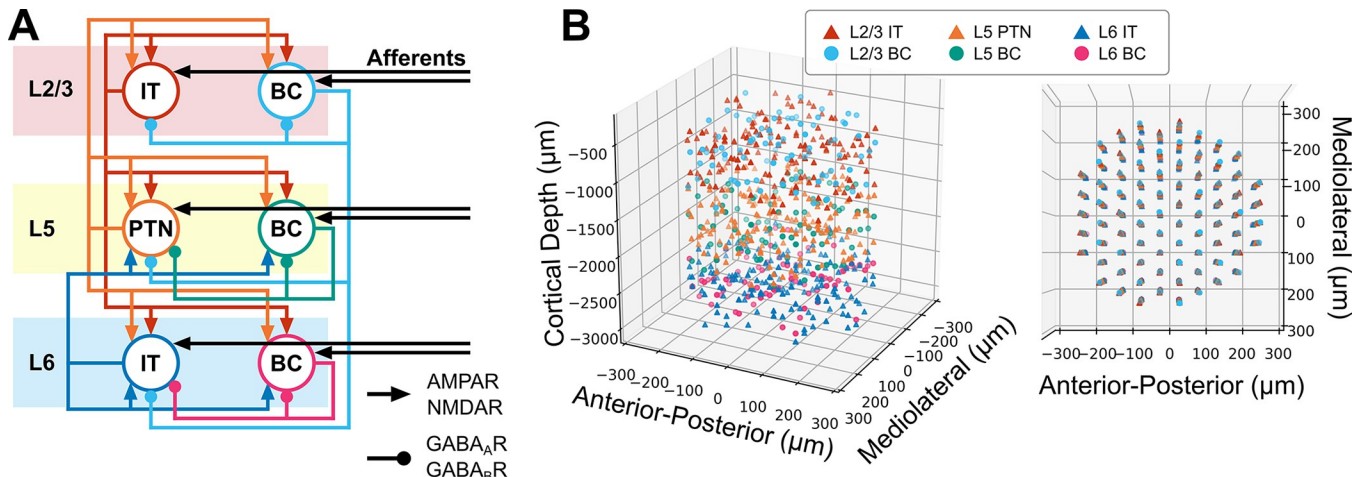

**Fig 3. Overview of motor cortical macrocolumn model.** A) Block diagram of cortical connectivity. Arrowheads denote excitatory connections mediated by AMPA and NMDA receptors. Round heads denote inhibitory connections mediated by GABA$_A$ and GABA$_B$ receptors. B) Three-dimensional representation of neuron locations. (Left) Side view showing laminar distribution. (Right) Top view depicting microcolumn organization within macrocolumn. IT: Intratelencephalic neuron. PTN: Pyramidal tract neuron. BC: Basket cell.

activation due to TMS was represented using an input–output approach. Given a stimulus intensity as input, the output was the proportion of the population that fired an action potential in response to the TMS pulse. Simulations were executed using NEURON 8.2.0+[10].

## Optimized models reproduce experimental data

Particle swarm optimization was used to identify parameters for a model capable of responding with a D-wave (D+) or without a D-wave (D-). The objective functions included the firing rate of the network prior to stimulation (i.e., no stimulation) and several properties of the corticospinal response after stimulation (see Methods for a detailed description of the experimental data) including the timings and amplitudes of the peaks and troughs. The parameters being optimized included the synaptic weights of each projection, the proportion of neurons activated by TMS, the conduction velocities for each neuron type, and the propagation delay due to stimulation of afferents. The total number of optimized parameters was 98, and the total list of parameters and their optimization ranges are described in Methods.

The final selected model had average corticospinal wave errors of 18.8% and 24.0% for the D+ and D− responses, respectively (Fig 4). The corticospinal tract activity generated by the individually optimized models captured many of the features of the experimental data (Fig 4A). The spiking responses of the models are represented using raster plots in Fig 4B. The final parameter values for each of the optimized models are presented in Figs A-C in S1 Appendix.

## Sensitivity analysis reveals parameters that preferentially contribute to corticospinal waves

Due to the high dimensionality of the parameter space (98 parameters), total grid search, random, or quasi-random sampling would require a prohibitively large number of simulations to characterize fully the relationships between the parameters and the corticospinal response. To reduce the computational cost, a two-variable-at-a-time (TVAT) sensitivity analysis was conducted. TVAT is a form of fixed-point analysis that varies two parameters simultaneously in a grid-search with the remaining parameters fixed at their original values. TVAT analysis is more computationally intensive than the widely used one-variable-at-a-time method, but allows characterization of pairwise interactions between variables [11,12].

TVAT analysis was performed using direct activation parameters and synaptic weights. All unique parameter pairs were varied in a grid search spanning the entire parameter range used in the optimization. The amplitudes of the simulated corticospinal waves were measured to construct amplitude maps as a function of the parameter pair involved, and polynomial regressions were used to characterize the amplitude maps. The total effect sizes, computed as the sum of effect sizes across all corticospinal waves, for the 20 most influential parameters are shown in Fig 5. For activation effects, activation of L5 PTNs (TMS-L5 PTN) had the largest effect size followed by activation of afferents to L5 PTN (TMS-L5 PTN AFF) and activation of L2/3 ITs (TMS-L2/3 IT). This is followed by activations of L6 BCs and ITs (TMS-L6 BC and TMS-L6 IT). Important projections included the L5 PTN-L2/3 IT, L2/3 BC-L2/3 IT, L5 PTN-L5 PTN, L2/3 IT-L5 PTN, and L5 BC-L5 PTN. All effect sizes are shown in Fig F in S1 Appendix.

The effect sizes of the parameters on each individual corticospinal wave relative to the total are summarized in Fig 5B. This plot reveals that while activation of L5 PTNs substantially affected D-waves, this parameter made minimal contributions to I-waves. The activation of afferents to L5 PTNs, L2/3 IT, and L6 IT most substantially affected the I1-wave. This analysis led to a subsequent grouping of parameters that preferentially influenced a single corticospinal wave versus parameters that affected multiple waves.

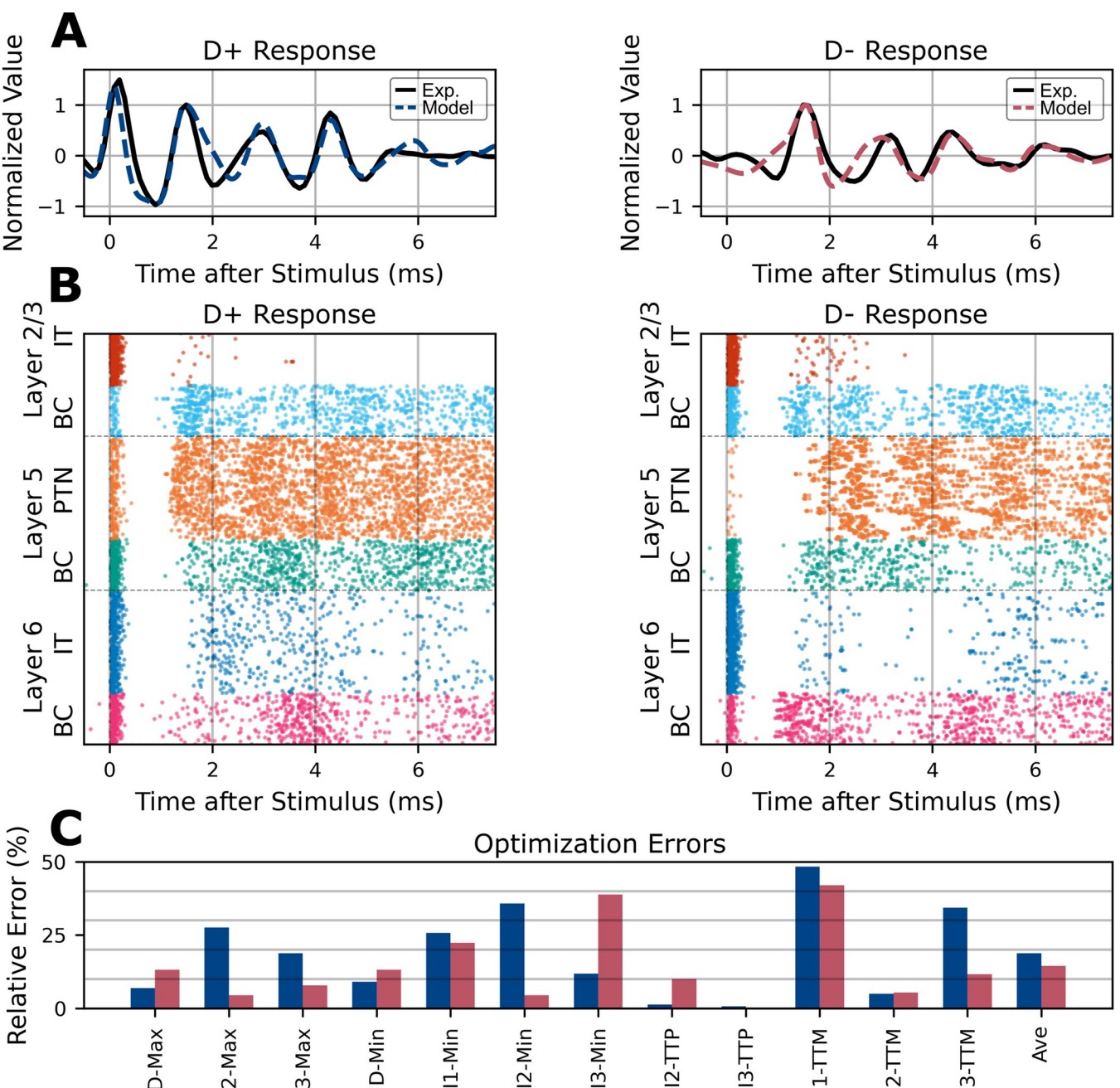

**Fig 4. Optimization results and unified model.** A) Simulated epidural corticospinal activity for the optimized model (dashed colored lines) compared to experimental data (solid black line). The same underlying cortical model parameters were used for the D+ response (left) and D- response (right) except for the TMS activation parameters which activated different proportions of the cell populations to produce each response. B) Spike raster plots for all motor cortical neuron types. A band-pass filter was applied to the activity of the Layer 5 PTN (orange) to represent the corticospinal responses shown in A. C) Distribution of relative errors across corticospinal wave objectives. Average error is plotted on the right side.

Different groupings of the total effect sizes were made to compare the average effect sizes of broader categories. The effect sizes were further subdivided based on corticospinal wave to quantify the sensitivity of the waves to the different groupings. Corticospinal waves were more sensitive to changes in activation vs changes in synaptic strength (Fig 6A). Sensitivity was greater for activation of motor cortical neurons than activation of extracortical afferents

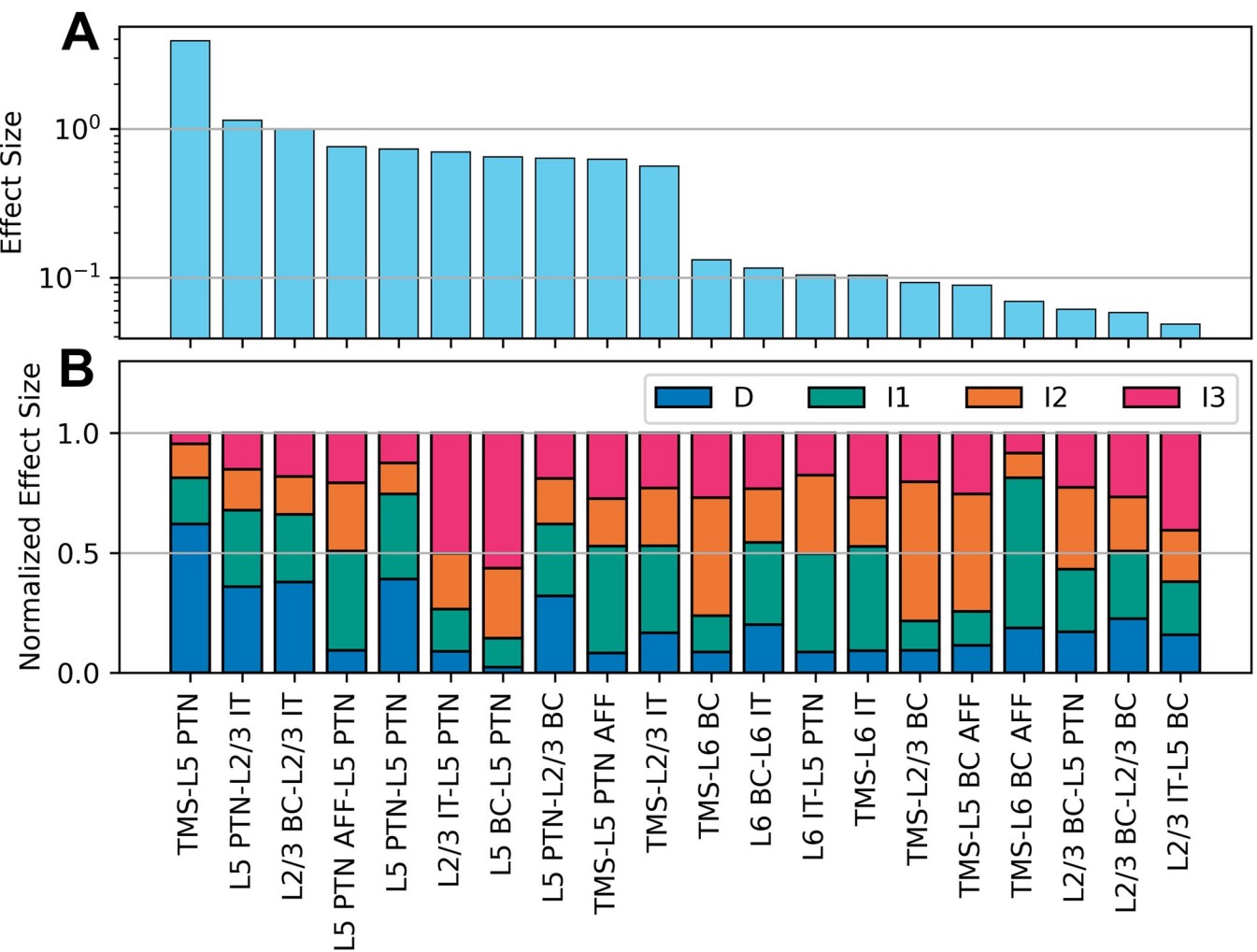

**Fig 5. TVAT effect sizes and their relative contributions across corticospinal waves.** A) Rank sorted total effect sizes across all waves are shown. Only the 20 largest effect sizes are shown for legibility; the full results are shown in Fig F in S1 Appendix. The y-axis uses a log-scale. B) Relative effect sizes normalized across all waves by parameter. A and B share the same x-axis. Parameter names were shortened and hyphenated such that the label before the hyphen corresponds to the presynaptic source and the label after the hyphen corresponds to the postsynaptic target, e.g. TMS-L6 BC indicates the activation of L6 basket cells via TMS and L2/3 BC-L5 PTN indicates the projection of L2/3 basket cells to L5 pyramidal tract neurons. IT: Intratelencephalic neuron. PTN: Pyramidal tract neuron. BC: Basket cell. AFF: Afferent.

(Fig 6B). This was primarily driven to the large effect size of activating L5 PTNs. At the circuit level, the synaptic strengths of afferents vs motor cortical neurons had an overall similar effect (Fig 6C). Sensitivity was greater for excitatory neurons vs inhibitory neurons (Fig 6D). Sensitivity was greater for the activation of feedforward excitation circuits, i.e., afferents that targeted excitatory neurons (Fig 6E). The strong sensitivity of the I1-wave is due to the effect of activating L5 PTN afferents. This is similarly reflected to the sensitivity to the synaptic strengths of afferents (Fig 6F). Sensitivity to feedback activation, i.e., activation of motor cortical neurons, was much stronger for excitatory neurons (Fig 6G). However, we can observe that sensitivity to inhibitory interneurons was greater in the later I-waves, I2 and I3. This is consistent with the literature. Sensitivity to synaptic strengths of intracortical projections was relatively similar across excitatory and inhibitory neurons (Fig 6H).

Separating the effect sizes for each corticospinal wave revealed that the individual parameters could preferentially affect one wave over others (Fig 5B). A parameter was defined as

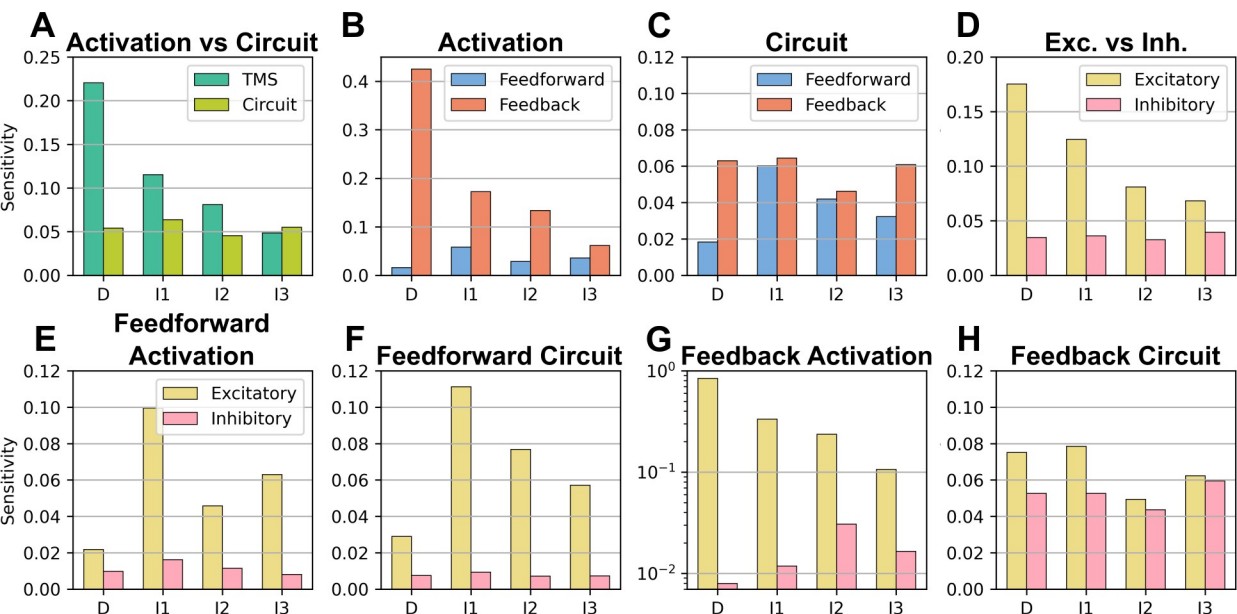

**Fig 6. Corticospinal sensitivities.** Sensitivity was computed as the average effect size for a specific corticospinal wave across all relevant mechanisms. A) Sensitivity was divided based on activation vs the synaptic strengths of the network. B) Sensitivity to activation was divided into feedforward activation, i.e., activation of extracortical afferent terminals, and feedback activation, i.e., activation of the motor cortical circuit. C) Sensitivity to the synaptic strengths was divided into the feedforward circuit, i.e., the synaptic strengths of extracortical afferents, vs the feedback circuit, i.e., the synaptic strengths of intracortical projections. D) Sensitivity was divided into elements that were excitatory vs inhibitory. E) Feedforward activation was divided into feedforward excitation, i.e., afferents targeting excitatory neurons, vs feedforward inhibition, i.e., afferents targeting inhibitory neurons. F) The synaptic strengths were divided into synaptic strengths of afferents targeting excitatory neurons vs synaptic strengths of afferents targeting inhibitory neurons. G) Feedback activation was divided into feedback excitation, i.e., activation of excitatory motor cortical neurons, vs feedback inhibition, i.e., activation of inhibitory motor cortical neurons. H) The synaptic strengths were divided into the strengths of excitatory intracortical projections vs inhibitory intracortical projections. Note the differences in y-axis values.

having a preferential effect if the parameter's largest effect size on a corticospinal wave was at least 50% larger than its second largest effect size. The activation parameters that preferentially affected each corticospinal wave were verified by visualizing the simulations performed for the TVAT analysis (Fig 7). These visualizations demonstrate that the sensitivity analysis was consistent with the actual simulations. The analysis identified that: the D-wave was most sensitive to the activation of L5 PTNs, the I1-wave was most sensitive to direct activation of afferents to L5 PTNs followed by activation of L2/3 IT and L6 IT, the I2-wave was most sensitive to activation of basket cells, and the I3-wave was most sensitive to activation of afferents to L2/3 ITs.

## Structural parameters that determine preferential influence

The sensitivity analysis predicted that multiple parameters could preferentially influence each I-wave. To identify any shared features that may predict preferential influence on the same corticospinal wave, a secondary analysis was conducted (Fig 8). The anatomical properties of the macrocolumn, such as the distances between neurons and connection probabilities, remained invariant during optimization. These invariant properties were quantified using a graph theoretical analysis, and machine learning was used to identify patterns in the network structure that contributed to corticospinal wave generation. Because only the L5 PTNs contributed to the signal recorded in the corticospinal tract, the relationships between neuron types to the L5 PTNs were characterized by deconstructing the network graph into simple paths, i.e., paths with non-repeating nodes. All directed simple paths for all neuron types leading to L5 PTNs were characterized for analysis. See Methods for detailed descriptions of the graph characterizations.

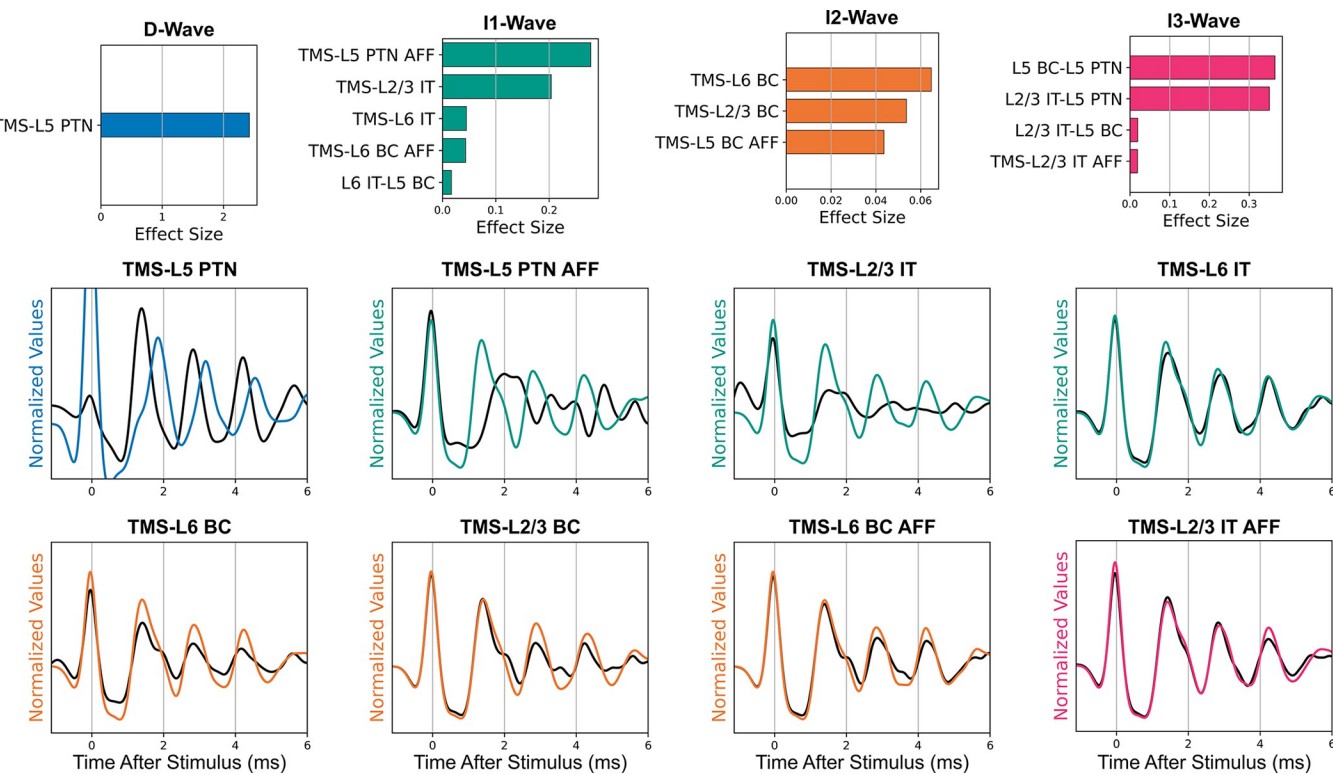

**Fig 7. Effect sizes for parameters that preferentially affected a single I-wave.** For each corticospinal wave, effect sizes for parameters that preferentially affected the wave were normalized and rank sorted and visualized as bar plots. Examples of traces that demonstrate the preferential effects of the identified activations are shown. The solid black line represents responses for which the parameter was set to zero. The difference in the amplitude of the wave across the colored and black lines indicates that the parameter was important to the generation of that wave. The waves are labelled in the plots. Please note the difference in x-axis limits across the bar plots. IT: Intratelencephalic neuron. PTN: Pyramidal tract neuron. BC: Basket cell. AFF: Afferent.

Recursive feature elimination was used to identify the features with the best classification performance (Fig 8A and 8C). The most important features to identify parameters that preferentially activated a single corticospinal wave were a strong average connection probability to L5 PTNs and whether the overall effect on the L5 PTNs was excitatory or inhibitory with a validation classification accuracy of 97.1% (Fig 8B). The most important feature to identify which corticospinal wave a preferential parameter affected was the conduction delay of the shortest path between the starting neuron and the L5 PTNs, and the validation accuracy was 81.6% (Fig 8C and 8D).

Although the sensitivity analysis identified important circuit mechanisms (i.e., activations and projections) involved in corticospinal wave generation, the subsequent machine learning analysis identified the anatomical bases that explained how and why the circuit mechanisms had a preferential effect. This secondary structural analysis provides a method for identifying fundamental principles involved in the neural response to acute stimulation.

## Discussion

We developed a data-driven model of a human motor cortical macrocolumn that generated realistic D-waves and I-waves in response to single pulse TMS. The unified model reproduced responses that included or excluded a D-wave primarily by changing the direct activation of L5 PTNs, which is consistent with the mechanisms of D-wave generation [4]. Other differences in activation (Fig A in S1 Appendix) to generate a D+ or D- response were necessary to

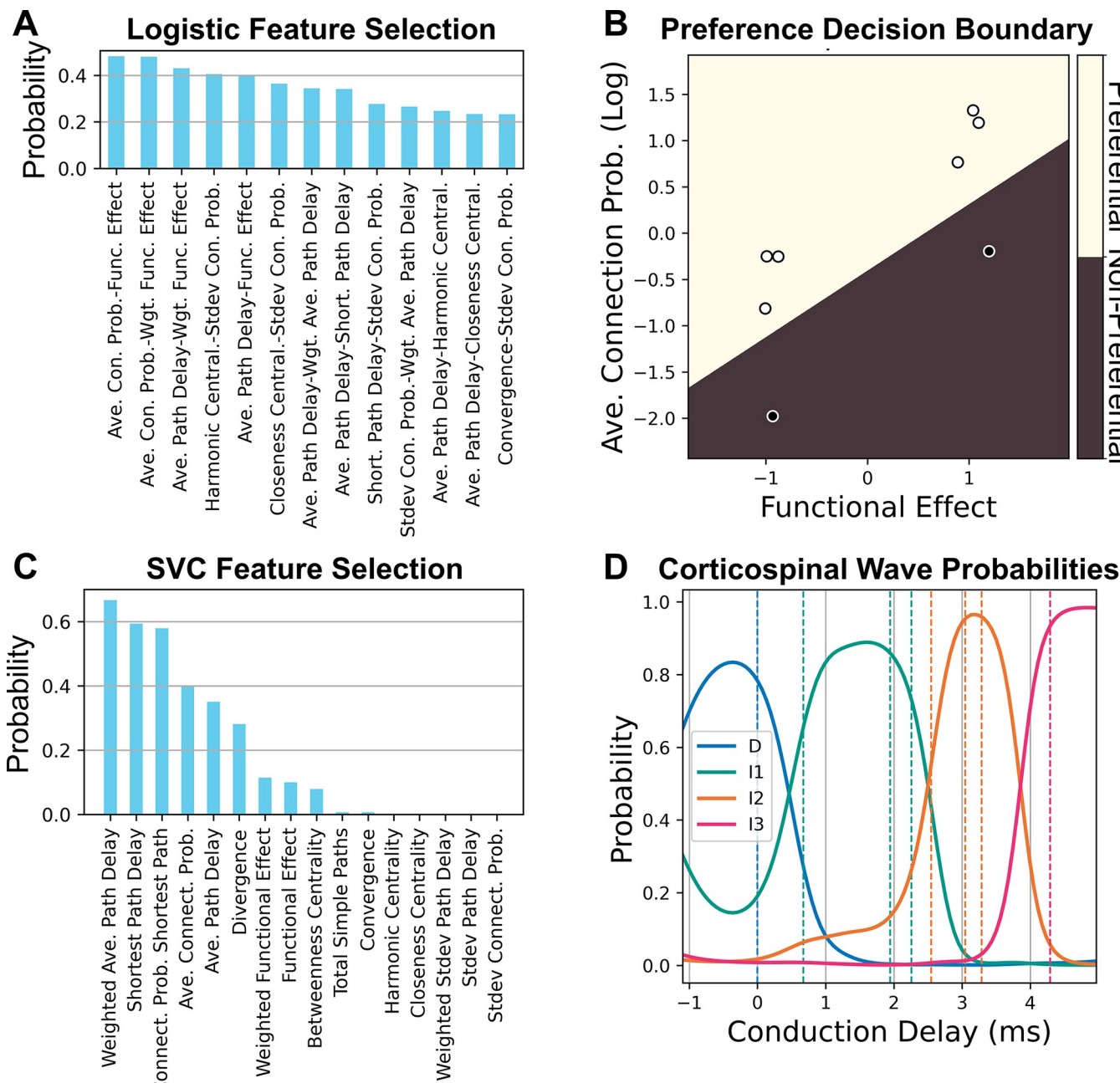

**Fig 8. Classification of network features for effect size types.** A) Recursive feature elimination was conducted to identify feature pairs that could predict preferential activation of corticospinal waves. Higher probabilities of remaining after elimination indicated better classification accuracy. Only a partial number of pairs are shown for legibility. B) Logistic regression decision boundary for preferential parameters (light) versus non-preferential parameters (dark) using the best classification features identified in A. Dark filled dots indicate data that were preferential, and light filled dots indicate data that were not preferential. C) Recursive feature elimination to identify features that predict corticospinal wave preference for preferential parameters. Only 10 features are shown for legibility. D) Corticospinal wave probabilities obtained by support vector classification using the single best classification feature from C. The dashed lines represent the conduction delays of the data being classified.

compensate for the transsynaptic effects on subsequent I-waves that were caused by a D-wave or reproduce the desired I-wave amplitudes without a D-wave. The synaptic strength of intra-cortical projections had multiple effects on the model, including changing the steady-state properties of the network prior to receiving the TMS stimuli. Few projections could

preferentially affect single corticospinal waves (Fig 7), and most affected multiple corticospinal waves due to their effect on the network's prestimulus, steady-state firing rates(Fig E in S1 Appendix). Furthermore, the final conduction velocities of the model were within experimentally reported ranges (Fig B in S1 Appendix). TVAT sensitivity analysis, which lies between a local and global sensitivity analysis, identified the circuit pathways and TMS activations important to I-wave generation.

The results of the sensitivity analysis support the hypothesis that direct activation of the terminals of afferents to motor cortex are an important mechanism for I-wave generation but are not consistent with the hypothesis that I-waves are generated by repetitive firing of single neurons (neural oscillator hypothesis). The analysis also supports the involvement of both excitatory and inhibitory neuron types in modulating I-waves [5]. In addition, the sensitivity analysis identified afferents and neuron types endogenous to the motor cortex that can be directly activated to generate corticospinal waves. Subsequently, structural analysis identified general structural principles that allowed these activations to preferentially generate corticospinal waves. Direct activation of afferents and neuron types can preferentially contribute to single I-waves if they have a highly connected path to L5 PTNs, relative to all other paths between the activated neuron type and L5 PTNs. Finally, the latency of the I-wave that is affected by a path can be predicted by its total conduction delay to L5 PTNs.

Several hypotheses have been proposed to explain I-wave generation and can be grouped into mechanisms at the network level vs the single neuron level [6]. Network level hypotheses propose either a single pathway of activation that simultaneously recruits multiple excitatory neuron populations to produce I-waves or multiple pathways of activation that recruit different excitatory neuron populations to produce different I-waves. Both hypotheses have a second variant that includes inhibitory neurons. The sensitivity analysis supports the hypothesis that multiple pathways are recruited to produce different I-waves and that interneurons serve an important role.

These network level hypotheses focused on I-wave generation being driven by afferents to the motor cortex, e.g., cortico-cortical fibers originating in premotor or somatosensory areas or fibers from thalamus. However, in addition to supporting the large effect sizes of afferents on I-waves, the sensitivity analysis identified mechanisms of I-wave generation that were endogenous to the motor cortical circuit, i.e., activation of intracortical motor cortex projections can generate I-waves. This is a novel hypothesis produced by the computational analysis of this work.

At the single neuron level, there are two major hypotheses. One is the concept of L5 PTNs as neural oscillators that burst during activation to produce I-waves. Our results do not support this hypothesis as the L5 PTN models tended to fire a single time during the course of the I-waves (Fig 9). These simulation results are consistent with recordings of single corticospinal axons in response to TMS [13]. The second hypothesis involves a mechanism involving calcium and the backpropagating action potential, but this hypothesis cannot be evaluated using our framework because the neuron models lack dendrites.

## Separate pathways for activation that include excitatory and inhibitory neurons

The leading hypothesis for I-wave generation proposes that 1) separate activation pathways exist for early versus late I-waves, and 2) activated pathways include both excitatory and inhibitory neurons [6,14]. The sensitivity analysis identified neural activations that preferentially modulated specific I-waves, revealed preferential activation pathways for all three I-waves, and showed that silencing their activation greatly suppressed a particular I-wave (Fig 7). The

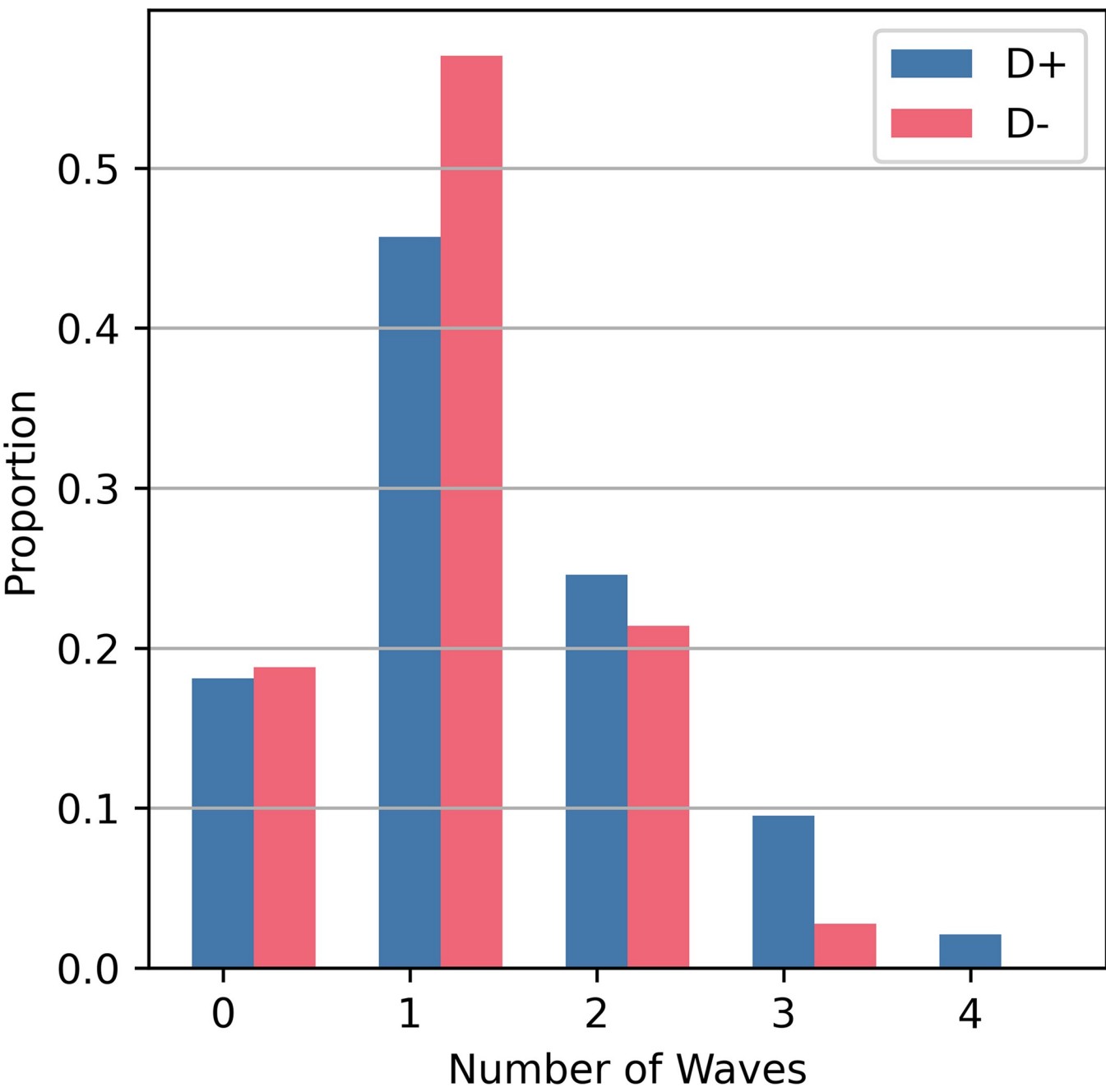

**Fig 9. Histogram of number of waves for which L5 PTNs contributed a spike.** For each stimulus presentation, the spikes generated by each L5 PTN were divided based on the time windows for each corticospinal wave, and the total number of time windows during which spiking occurred was counted.

sensitivity analysis was grouped to compare the total effect sizes of excitatory and inhibitory neurons on I-wave generation and revealed that corticospinal waves exhibited comparable sensitivities to both excitatory and inhibitory neurons and that inhibitory neurons are involved in I-wave modulation (Fig 6H).

Most inhibitory neurons had non-preferential effects, i.e., affected multiple I-waves, which is consistent with experimental findings that various anesthetics, which act as allosteric modulators of $GABA_A R$, generally reduce I-wave amplitudes [6]. Additionally, the sensitivity

analysis showed that the I2-wave and I3-waves were most sensitive to activation of inhibitory neurons (Fig 5G), and this is consistent with experimental findings that show GABA$_A$ agonists affect later I-waves [15–17].

### Direct activations of the endogenous circuit contribute to I-waves

The prior conceptual frameworks assumed that I-waves are initiated by activation of cortico-cortical fiber afferents, and the sensitivity analysis supports that the corticospinal response is most sensitive to activation of terminals of afferents. However, this analysis revealed that activation of the motor cortical circuit itself can initiate I-waves. For example, activation of ITs in L2/3 and L6 preferentially activated I1-waves (Figs 5 and 7). Although designing *in vivo* TMS experiments that control for contributions of endogenous circuit elements to I-waves is difficult, the modeling results suggest activation of the endogenous circuit as another mechanism for I-wave generation, in addition to activation of afferents. Intracortical microstimulation (ICMS) studies can provide some insight into intracortical TMS effects and are further discussed below in the Comparison to Intracortical Microstimulation subsection.

### L5 PTNs as population oscillators but not neural oscillators

Another category of hypotheses for I-wave generation is the concept of the neural oscillator. These theories were motivated by the fact that L5 PTNs can achieve firing rates that match the frequency of I-waves and led to exploration of cellular mechanisms for I-wave generation [6]. A histogram was constructed of the spike counts for each L5 PTN during the different I-waves (Fig 9), and L5 PTNs were most likely to contribute to a single I-wave during the corticospinal response. This is consistent with observations from [13] that reported that neurons tend to fire a single time during the corticospinal response to TMS. However, at the population level excitatory recurrent connections exist between L5 PTNs, and the sensitivity analysis demonstrated that the recurrent connections are involved in I-wave modulation as seen in Fig 5C. Therefore, the modeling results do not support that I-waves are generated or sustained at the neuronal level; rather, their generation appears to be a population level effect.

### Connectivity and conduction delay as mechanisms for preferential I-wave generation

Given that multiple mechanisms can preferentially contribute to the same I-wave, the structural analysis sought to identify the commonalities among mechanisms that yielded this response. A neuron type within the circuit could have multiple paths leading to L5 PTN with different properties for each path. Neuron types with a single path that had a high connection probability to L5 PTNs, relative to other paths starting from the same neuron type, could preferentially affect a single I-wave (Fig 8A and 8B). For neuron types where such a path exists, the primary mechanism for determining early versus late I-wave activation was the conduction delay of the path between the activated population and L5 PTNs (Fig 8C and 8D). The conduction delay defined in this study represents the combined contributions of action potential propagation along the axon, synaptic transmission, and somatodendritic propagation of the resulting postsynaptic potential. This is supported by the computational work of Rusu and colleagues who controlled conduction delay based on synaptic location within dendrites [8].

To generalize, the results of the structural analysis suggest that if the generator of a signal within a network is known, and the connection probabilities and conduction delays of the network are known, then the network elements that preferentially contribute to singular peaks of a system's impulse response can be screened by performing the following: for each neuron type 1) identify all possible paths from the neuron type to the signal generator, 2) compute the

ratios of the log of the connection probability between the most highly connected path and the remaining paths normalized by the sum of all log probabilities, and 3) obtain the latency of effect for the most highly connected path. Neuron types that have a path that is more highly connected than the remaining paths will have a preferential influence on peaks that occur during their latency of effect.

## Comparison to Intracortical Microstimulation (ICMS) Studies

Direct cortical recordings to investigate I-waves are currently limited due to the technical challenges of suppressing the TMS artifact, which saturates recordings and prevents recovery of the activity during the period when the D-wave and I-waves occur [18,19]. ICMS in animals can generate high frequency multiunit activity with frequencies comparable to I-waves [20–22]. The results of ICMS studies can contribute to understanding the TMS response, but due to the differences in the spatial distribution and gradient of the electric field, ICMS studies cannot be used to explain fully TMS evoked I-waves [23].

ICMS applied to the primary motor cortex (M1) hand area in nonhuman primates showed that earlier peaks were elicited if the stimulation was closer to the recording site [22]. The study hypothesized that the stimuli were activating horizontal fibers within M1, and these results support conduction delay as a mechanism determining the latencies of peaks. The horizontal fibers further represent afferents, relative to a macrocolumn, that are endogenous to M1. Single unit activity from a similar ICMS study that stimulated and recorded from M1 found minimal, sparse spiking within the time window relevant for I-waves and supports that single L5 PTNs contribute to few I-waves, if at all [20]. This corroborates the modeling predictions that I-waves represent a population response comprised of heterogeneous, sparse spiking rather than a synchronized rapid spiking response across neurons (Fig H in S1 Appendix). Another ICMS study stimulated a region of the ventral premotor area F5 that sends afferents to the hand knob area of M1 [21]. Stimulation of F5 at lower intensities recruited the I1-wave first, and higher intensities eventually recruited later I-waves. Although it is known that F5 projects to M1, the laminar distribution of the terminals of F5 afferents in M1 are unknown. Nonetheless, these results are consistent with the modeling prediction that the I1-wave is most sensitive to activation of afferents. Maier and colleagues also stimulated M1 directly and found that D-waves are much less likely to be elicited than I1-waves. This finding is in line with the TMS literature [24], and the sensitivity analysis (Fig 6C) is also consistent with these experimental observations in that the I1-wave is most sensitive to stimulation of afferents compared to the D-wave, which is least sensitive.

## Putative afferents for I-wave generation

In the present model, afferents were represented as spiking inputs that were specific for each neuron type in the model, and the effect of TMS was represented by activation of the axon terminals of these afferents within the motor cortical macrocolumn. The sensitivity analysis predicted that activation of afferents for specific neuron types could have a preferential effect on specific I-waves, so the results of the sensitivity analysis were compared to the laminar distribution of terminals of corticocortical afferents in mouse motor cortex [25] to predict the anatomical origin of afferents with preferential I-wave effects. Afferents originating from the secondary (supplementary) motor area (M2) have a high density of terminals in the deep portion of L5 where the somata of L5 PTNs lie, and activation of M2 afferents may be a candidate for I1-wave generation. Afferents from the primary somatosensory cortex have a high density of axon terminals in L2/3 and superficial L5 and could be important for I1-wave generation. The axon terminals of the orbital cortex primarily target L6 and may contribute to I1-waves.

The axon terminal distributions for lateral and anterior ventral thalamus within motor cortex were also characterized [25], but prior studies showed that lesions in those areas do not affect I-wave generation [26].

The laminar distribution of horizontal connections between columns within motor cortex have not been directly characterized. However, Narayanan and colleagues reported the laminar distribution of axon terminals endogenous to rat primary somatosensory cortex [27]. The horizontal connections of L2/3 and L5 pyramidal neurons are most dense in L2/3, which may contribute to the I1- and I3—waves. The horizontal connections of L6 pyramidal neurons are most dense in deep L5 and L6 which may contribute to I1- and I2-waves.

## Model limitations and future directions

An important design criterion for the modeling work was computational efficiency to enable the parameter explorations necessary for optimization and sensitivity analysis. In general, computational gains came at the expense of biological details and constraints. However, the simplified model enabled more specific and in-depth computational analyses. To benchmark the difference, a Blue Brain L5 neuron model [28] with realistic dendrites was compared to the Esser L5 PTN point neuron model. Both were driven by identical 1000 Hz Poisson spike trains with synaptic weights adjusted to produce identical output firing rates (5 Hz). Simulating 10 s of time resulted in execution times of 1500 s for the Blue Brain model and 0.8 s for the Esser model. Hodgkin-Huxley style models with realistic morphologies could be up to 1900x slower than leaky-integrate-and-fire models.

However, point neuron representations precluded any analyses involving dendrites, axons, spatial integration of postsynaptic potentials, or ephaptic coupling. Spatially extended, i.e., morphologically realistic, neuron models, could accommodate these mechanisms and enable the exploration of their contributions to modulation of I-waves. Furthermore, this work represented TMS stimulation using an input–output approach, i.e., a given stimulus intensity resulted in some proportion of neurons of a particular type to fire an action potential. Therefore, the results cannot be generalized to explain effects to conditions beyond the experimental data it was optimized to reproduce, i.e., the results are only valid for coil orientations that induce an electric field in the posterior-anterior direction using a monophasic pulse. Without additional data, the results and model cannot be reliably extrapolated to responses in other orientations such as lateral-medial, for other stimulation waveforms such as biphasic pulse responses, or to generate I-waves beyond I3. More realistic representations of the electric field coupling could allow generalizations to other conditions by modeling the spatial distribution of activation of the induced electric field using finite element modeling [23,28–30]. Combining these methods with neuronal models with realistic morphologies [31–33] would yield informative insights.

The representation of extracortical afferents could also be improved. Afferents were represented as spiking processes that targeted specific neuron types. More realistic representations of afferents with distributions and connectivities that matched anatomical data would more directly address the effect of specific fibers on I-waves. Nonetheless, allowing afferents to be separately variable for each neuron type provided a basis to understand their contributions.

Adding more neuron types is also necessary to include more types of circuits for analysis. Traditionally, L4 in motor cortex has been described as either nonexistent or very thin, which led motor cortex models to exclude L4 or represent it with inhibitory neurons only [7,34,8]. Recent evidence has identified excitatory IT neurons in L4 with projections to L2/3 [35–37] leading to more complex models of M1 [33]. The present modeling results predict that, while not included, L4 IT neurons would participate in later I-waves due to their strong projection

into L2/3. Interneuron types such as SOM+, VIP+, and CCK+ would also enrich the network and allow a holistic analysis of the network response.

Given the importance of conduction delays, expanding the volume of motor cortex may be important. This work modeled a single macrocolumn comprising multiple microcolumns. Communications across adjacent macrocolumns, i.e., intracortical afferents, could alter the corticospinal response to TMS as they represent "afferent" inputs to macrocolumns that arise within the motor cortex. Their interactions could further modulate I-waves through both excitatory and direct inhibitory projections, and the latencies of the feedback will likely cause adjacent macrocolumns to contribute toward late I-waves.

The analysis could also be improved by including more types of data. Experimental data from only two subjects was used with responses from a single TMS intensity. The data were representative of the two qualitative types of responses—with and without D-wave. The small dataset allowed for more rapid model development due to fewer optimization constraints, and the methods established in this work can be applied in the future to extended data from more subjects and more recordings within subject. Furthermore, the optimization included only a single stimulus intensity as a constraint. Incorporating corticospinal recordings in response to multiple stimulus intensities from the same subject could reveal differences in effect size or engaged mechanism as a function of intensity.

Additionally, the predictions from the model are limited to the single pulse response and are not readily extendable to paired pulse or repetitive pulse paradigms. This is partly due to $GABA_BR$ parameters being underconstrained. $GABA_BR$ conductance was partially constrained by the baseline firing rate objective but has been shown to have no effect on I-waves [38]. However, $GABA_BR$ is important for the cortical silent period [39] and paired pulse responses [40], and these data can be incorporated as optimization objectives in future work. Including more interneuron types such as somatostatin or vasoactive intestinal peptide expressing interneurons would further disambiguate contributions of both $GABA_AR$ and $GABA_BR$.

Finally, the model lacks a representation of short-term plasticity (STP), which contributes to non-linear facilitative and depressive effects at short-time scales. Though STP is engaged in response to paired-pulse stimuli [41], it is unknown the degree to which the series of transsynaptic activations resulting from a single pulse also contribute to I-wave, and remains an open question.

## Conclusions

To understand the mechanisms and principles underlying a biological process, sensitivity analysis is a powerful tool. However, as the number of relevant variables increases, the analysis can become overwhelming, and conclusions become diluted. At these large numbers, degeneracy in the sensitivity analysis is possible as many mechanisms can be identified to be significant to the phenomenon of interest. However, there is also the possibility that subsets of these mechanisms share certain properties that represent a more fundamental mechanism or at least a lower-level mechanism that was previously unclear or unaccounted for. In this case, a secondary analysis can reveal these lower-level mechanisms that underlie the variables that explain the phenomenon of interest.

In this work, the sensitivity analysis supported one of the major hypotheses concerning I-wave generation: I-waves are recruited transsynaptically through separate circuits that impinge onto L5 PTNs and involve both excitatory and inhibitory neurons. Additionally, activation of afferents onto L5 PTNs and non-L5 ITs cells was important for I-wave generation. The secondary analysis revealed that the anatomical structure of the network, i.e., the wiring diagram

and conduction latencies that resulted from the anatomical constraints, were then important for predicting the circuit activations that give rise to specific I-waves with both the recruitment of afferents to L2/3 and L6 IT cells being possible mechanisms.

Finally, the lower-level nature of the mechanism identified using the secondary analysis allows these insights to be generalized beyond the motor cortex and TMS. Understanding the circuit organization of the target neural system and its inherent conduction latencies can be used to screen for important pathways that are recruited and contribute to an acute evoked response.

## Methods

### Ethics statement

Experimental data were obtained from human subjects who had spinal cord stimulators implanted to treat drug-resistant dorso-lumbar pain. Data was collected in accordance with an experimental protocol that was approved by the Ethics Committee of Campus Bio-Medico University of Rome with formal written consent obtained from the subjects. Use of the data in this study was approved by the Institutional Review Board of the Duke University Health System.

### Motor cortical column simulations

**Neuronal network model.**   The motor cortical macrocolumn model was based on the equations and parameters published by Esser et al., 2005, which specified the connectivity, somatic biophysics, and synaptic properties [7]. The model contained L2/3 ITs and BCs, L5 PTNs and BCs, L6 ITs and BCs and excitatory afferents that targeted each neuron type (i.e., six groups of afferents). The circuit describing the connectivity is shown in Fig 3A. The Esser model was chosen as a starting point due to its ability to generate I-waves and the low computational complexity of its leaky-integrate-and-fire, point neuron models. The spiking activities of the afferents were generated by a Poisson process with a mean firing rate of 0.25 Hz [42]. Noise was added to the neuron models that was independent of the synaptic drive provided by the afferents and unaffected by TMS to ensure proper baseline firing rates and reduce network synchronization. Each neuron received its own noise in the form of short, suprathreshold current injections with Poisson-distributed intervals. Although the Esser model included the thalamus and thalamocortical projections, the thalamus was omitted from the present work to further reduce computational time because it does not affect I-wave generation [26].

The macrocolumn encompassed a cylinder with a diameter of 500 μm (Fig 3B) based on anatomical studies [43]. The height of the cylinder was 2700 μm based on measurements made on human motor cortex from ex vivo brain [44]. This study also informed the total vertical thickness (i.e., depth) of the layers within the macrocolumn. The cortical depth location of a neuron was uniformly and randomly generated within the appropriate layer bounds. The macrocolumn was comprised of microcolumns that were arranged in a triangular lattice with a spacing of 50 μm [45] resulting in 79 microcolumns and matched the ratio of microcolumns per macrocolumn [43,46]. The microcolumns were synonymous with the "topographical elements" described in the Esser model and contained 2 excitatory neurons and 1 inhibitory neuron per layer. With 3 neurons per layer, 3 layers per microcolumn, and 79 microcolumns in the macrocolumn, there was a total of 711 neurons (Table 1).

The conduction delay, defined as the time between the onset of an action potential and the start of the postsynaptic potential at the soma of the postsynaptic neuron, was calculated from the distance between the presynaptic and postsynaptic neuron pair and conduction velocity.

**Table 1. Total numbers of neurons in model.**

| Neuron type | Number |
| --- | --- |
| L2/3 IT | 158 |
| L2/3 BC | 79 |
| L5 PTN | 158 |
| L5 BC | 79 |
| L6 IT | 158 |
| L6 BC | 79 |
| L2/3 IT AFF | 79 |
| L2/3 BC AFF | 79 |
| L5 PTN AFF | 79 |
| L5 BC AFF | 79 |
| L6 IT AFF | 79 |
| L6 BC AFF | 79 |

The conduction velocity measured from non-human primates (0.570 m/s) was used as human measurements were not available [47].

TMS activation included only suprathreshold effects. Based on the specified activated proportion for the chosen afferent or neuron type, a corresponding proportion of that given population or afferent type was randomly chosen to be activated, and neurons/afferents were randomly selected for each presentation of the stimulus. Direct activation of neurons resulted in an injection of a short suprathreshold current to elicit an action potential that was propagated orthodromically to all postsynaptically connected neurons using all relevant conduction delays. Direct activation of the terminals of afferents resulted in the activation of all connected synapses with the appropriate conduction delays. Though studies have demonstrated that motor thresholds in response to TMS are lower in presynaptic terminals [29], other studies have showed the primary axon to have a large influence on the sensitivity of a cortical pyramidal cells to TMS [48]. We found that antidromically propagated action potentials result in conduction delays that are similar to orthodromically propagated action potentials (Fig G in S1 Appendix). Because this study used implicit, functional representations of axons through conduction delays, we found this strategy for activation to be representative of the current theories of activation.

Connectivity parameters, neuron parameters, and synaptic parameters were identical to those reported in [7] with the following exceptions. Orientation selectivity-based connectivity was not included, so microcolumns could connect to any of their neighbors rather than being restricted to microcolumns with similar orientation sensitivity. Because the geometric area of the model was reduced from the original, the overall synaptic drive was decreased, and the subsequent optimization allowed larger synaptic weights to compensate.

**Simulation paradigm.** Simulations were designed to ensure that the network achieved steady-state before firing rates were measured, and steady-state properties were measured between 500 and 2000 ms. To reduce synchronization of the network due to simultaneous activation of afferent inputs, the onsets of the Poisson spike trains of the afferents were randomly and uniformly selected between 0 and 200 ms. TMS stimuli were applied at 2000 ms with inter-trial intervals of 200 ms with a total of five trials. This interval was selected based on population averages of trials, which showed no longer-term effects beyond 150 ms. Furthermore, the model did not include synaptic plasticity or thalamic connections. Analysis of the TMS response was conducted on the trial average. The total simulated time was 3000 ms and had an execution time of 49 s.

Simulations and analyses were run on the Duke Compute Cluster on nodes comprised of a heterogeneous mix of hardware including the Intel Xeon CPU E5-2680 v4 @ 2.40GHz, Intel Xeon CPU E5-2680 v3 @ 2.50GHz, Intel Xeon Gold 6154 CPU @ 3.00GHz, Intel Xeon Gold 6254 CPU @ 3.10GHz.

**Selecting an appropriate time-step.**   The time-step was decreased from the value originally used in Esser et al., 2005, from 0.1 ms to 0.01 ms due to instabilities in the network during these longer simulations [7]. The time-step was selected by running single neuron simulations while log-linearly varying the time-step from 0.001 to 0.2 ms. The models received a random Poisson input with a mean firing rate of 1000 Hz for 20 s of simulated time. This firing rate was representative of the total firing rate across all inputs that a neuron would experience during a simulation used to evaluate the model response to TMS. The response at 0.001 ms was used as the baseline response, and the model behavior were characterized using the following metrics: Number of spikes generated, mean inter-spike interval (ISI), coefficient of variation of the ISI, normalized root mean square error (NRMSE) of the membrane potential, and the van Rossum spike distance [49].

The van Rossum spike distance was computed by convolving two spike trains using a causal exponential kernel and computing their L2 norm following equation

$$spike\ distance = \sqrt{\frac{2}{\tau} \int_0^\infty \left[h(t;\ u) - h(t;v)\right]^2 dt}$$

where $\tau$ is the time constant of the causal exponential kernel, $^2/_\tau$ is a normalizing factor, $h(t)$ is the kernel function, and $u$ and $v$ are the two spike trains. A time constant of 500 ms was used for the spike distance because the 0.001 ms time-step case had a mean ISI of approximately 500 ms.

For each time-step, 50 simulations/trials were conducted. Each trial used a different random seed to change the Poisson input, and the sequence of random seeds for the trials was identical across time-steps. The mean of the metrics across trials for each time-step was calculated for further analysis.

Most metrics did not appear to have arrived at an asymptote as the time-step was reduced to 0.001 ms (Fig 10). However, for the number of spikes, mean ISI, and the coefficient of variation of the ISI, an asymptote was reached at approximately 0.01 ms. Thus, the 0.01 ms time-step was selected.

## Experimental data

The experimental setup is summarized in Fig 1A. For each subject, an electrode array was implanted percutaneously in the cervical epidural space, with the recording sites aligned vertically along the dorsum of the cord. Spinal potentials were recorded differentially between proximal-distal pairs of contacts (with the distal contact connected to the reference input of the amplifier), amplified and filtered (gain: 10000; bandwidth: 3 Hz to 3 kHz) by a Digitimer D360 amplifier (Digitimer Ltd., Welwyn Garden City, UK), and sampled at 10 kHz by means of a CED 1401 A/D converter (Cambridge Electronic Design Ltd., Cambridge, UK).

A figure-of-eight coil with external loop diameter of 70 mm was held over the right motor cortex at the location at which the threshold to elicit motor evoked potentials measured at the first dorsal interosseous (FDI) was with the induced current flowing in a posterior–anterior direction across the central sulcus. Monophasic pulses were delivered with a Magstim 200$^2$ stimulator (The Magstim Company Ltd., Whitland, UK), once every 5 seconds. Pulses had a rise time of 100 μs and a duration of 1 ms.

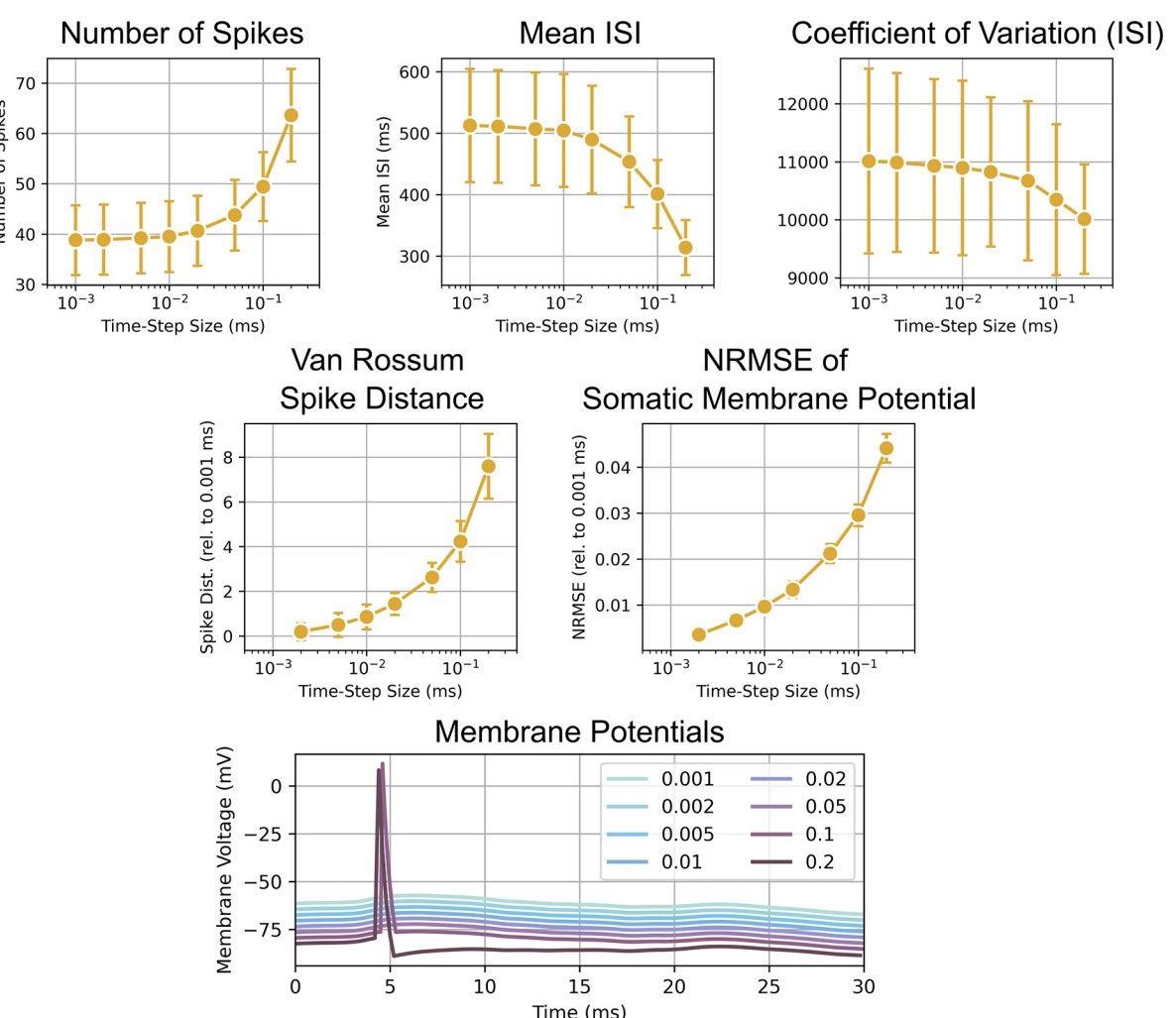

**Fig 10. Analysis to select a time-step that both minimizes computation time and is numerically stable.** Each scatter-line plots shows the mean of a metric as a function of the time-step size in the $\log_{10}$ scale. At the bottom, the membrane potentials of the neuron model for different time-steps are shown. Offsets were added for the y-axis to allow all lines to be distinctly seen. The plots depict a key behavior that differentiates simulations at larger time steps. A pronounced afterhyperpolarization is seen with a 0.2 ms time-step that is absent from other time-steps. Additionally, spikes are generated at larger time-steps (0.1 and 0.2 ms) that are absent for smaller time-steps. These dynamics contribute to the larger numbers of spikes, lower mean ISIs, larger NRMSE, and larger spike distance observed for larger time-steps.

Two subjects were included in this study (Fig 1B) with the stimulator output set to 120% of their respective resting motor thresholds (RMT). Subject 1 was female, 64 years old, and had a cervical epidural electrode implanted at C3–C5 level; the RMT of TMS was 34% of maximum stimulator output. Subject 2 was male, 68 years old, and had a cervical epidural electrode implanted at C1–C2 level; the RMT was 55% of maximum stimulator output. Subject 1 did not exhibit a D-wave in response to TMS (D-), while Subject 2 exhibited a D-wave (D+). These subjects were selected to investigate the mechanisms that underly these differences in the evoked response, given the same RMT intensity. Each subject received at least 30 pulses. For analysis, the responses were truncated to begin 2 ms after the TMS pulse to remove stimulation artifact. An additional noncausal bandpass filter (second-order Butterworth, 200 Hz to 1500 Hz) was applied to remove residual stimulus artifact, potential motor artifacts, and higher frequency activity that is unrelated to the corticospinal waves. Measurements of the corticospinal

response were taken from the filtered, trial-averaged signal. The differences in recording locations resulted in an average delay of 0.2 ms for the D- subject who was recorded at C3-C5 relative to the D+ subject who was recorded at C1-C2. These delays were appropriate based on estimates of corticospinal conduction velocity [13] and path distances from motor cortex to C1-C5 [50,51].

## Optimization of network model

**Particle swarm optimization.** Particle swarm optimization (PSO) is a metaheuristic algorithm for parameter exploration with the goal of finding parameters that satisfy one or more objectives. The particle's position represents the parameter values for the model, and a velocity term updates the position using a weighted combination of the best solution found by the particle itself (cognitive best) and the best solution found among a particle's neighbors (social best). PSO was implemented by modifying the *inspyred* Python software package [52].

Neighborhoods were constructed using a star topology with each particle's neighborhood size being 5% of the total number of particles. Optimization used 499 particles and ran for 300 iterations before termination. The optimization was initially repeated for each model five times to increase coverage of the parameter space and the likelihood of locating a global best solution. A best model was then selected, and a subsequent regularized optimization was repeated five times to identify a regularized model. Optimization evaluated particles in parallel across 499 CPUs while using a single main CPU to collect, analyze, and update particle positions. Each iteration took an average of 444 s, which is greater than the execution time of a single simulation, but the communication overhead, file i/o, and analysis after each iteration of particle evaluation added extra time. Each optimization had an average execution time of 37 hours, or 18,500 compute-hours. The two subject-specific models had ten total optimization runs using a total of 370,000 compute-hours to complete. Optimizations utilized an average of 131.21 GB of RAM.

At the beginning of the optimization procedure, particle positions were initialized using Sobol sampling. Sobol sampling generates a low-dispersion quasi Monte-Carlo sequence that exhibits better coverage of the parameter space than uniform random sampling for high-dimensional spaces and has been shown to improve optimization convergence [53].

Particle behavior was guided by inertial velocity, cognitive velocity, social velocity, gain factor, and noise [54]. Inertial weight corresponded to a particle's resistance to movement and results in a particle moving towards its previous position. The cognitive weight determined a particle's preference towards the position of the best solution it had found. The social weight determined a particle's preference towards the position of the best solution its neighborhood had found. The cognitive and social velocities were also separately modified using scalars drawn from a uniform distribution between 0 and 1. The velocity was then computed as the weighted average using the inertial, cognitive, and social weights. Finally, the velocity was scaled by the gain factor. For each particle coordinate, noise was sampled from a zero-mean Gaussian distribution with the standard deviation controlling the strength of the noise. Optimization noise is also known as mutation and was shown to be necessary for theoretical global convergence of PSOs [55]. Finally, the particle position was updated using both velocity and noise.

These optimization parameters were updated during optimization to switch from an initial stage of exploration to a final stage of convergence (Fig 11). During exploration, inertial weight, cognitive weight, gain factor, and noise were high, and the social weight was low. During convergence, the social weight was high, and the remaining terms were low. The

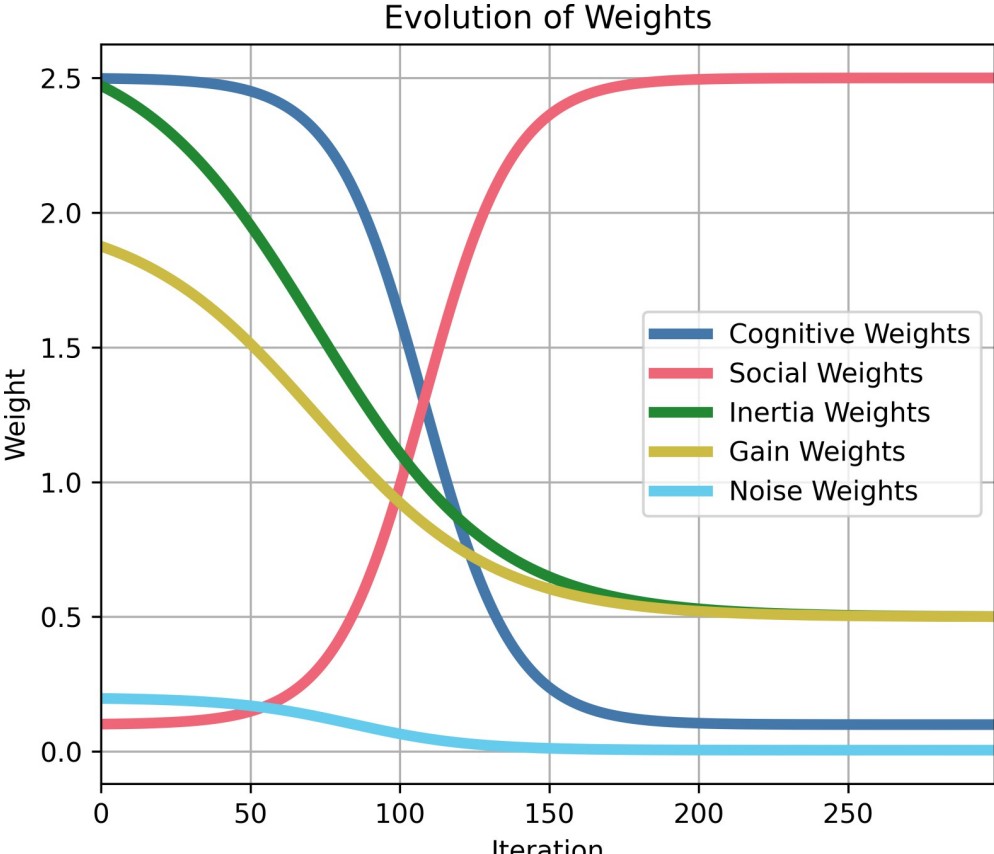

**Fig 11. Change in particle swarm optimization weights across successive iterations.** For approximately 100 iterations, optimization is exploratory with large cognitive, inertial, and gain weights before favoring convergence with high social weights for the final 150 iterations.

progression of the parameters followed a sigmoidal function

$$y(x) = A + \frac{K}{1 + e^{(ax - bN)/N}}$$

where $x$ is the current iteration of the optimization, $N$ is the total number of iterations for the algorithm, $A$ is the offset, $K$ is the amplitude and direction of the sigmoid, $a$ controls the steepness of the transition, and $b$ controls the midpoint of the transition. The parameters for the sigmoidal function are reported in Table 2.

A damped, reflecting boundary condition was implemented on the parameter search space [56]. If a particle's position exceeded a boundary, then the particle was reflected back into the valid parameter space using the difference between the original, non-valid position and the boundary. The reflection was damped by multiplying the difference with a scalar sampled from a uniform distribution between 0 and 1.

$$x_{reflect} = bound - U(0,1) * (x_{new} - bound)$$

**Optimization objectives.** There were four main groups of objectives: baseline activity, TMS response, synchrony, and miscellaneous. The miscellaneous group included objectives

**Table 2. Sigmoid function constants underlying evolution of optimization hyperparameters.**

| Parameter | A (Minimum) | K (Amplitude/Direction) | a (Slope) | b (midpoint) |
|---|---|---|---|---|
| Cognitive Weight | 2.5 | -2.4 | 20 | 7.2 |
| Social Weight | 0.1 | 2.4 | 20 | 7.2 |
| Inertial Weight | 0.5 | 2 | 15 | 4.2 |
| Gain Weight | 0.5 | 1.5 | 10 | 2.4 |
| Noise Weight | 0.005 | 0.195 | 15 | 4.2 |

that didn't fall into the previous groups but were also not well-related to each other. However, this lumping was necessary to reduce the dimensionality of the pareto front for visualization. The relative error was computed for each objective except when the objective was zero, in which case the absolute error was computed. The sum of the relative and absolute errors was used to represent the total error of a particle. Table 3 lists all objectives.

The baseline activity objectives included both the mean population inter-spike interval (ISI) and the mean population firing rate for the different neuron types. Both objectives were important to constrain the network activity due to the nature of their calculations. Firing rate was evaluated as the number of spikes elicited within a time-window. However, there was a possibility that the ISIs within the window were very small due to bursting behavior. Therefore, the mean ISI was added as an additional objective. Mean ISI alone was not a good objective for overall activity because the calculation of relative error resulted in lower error for small ISIs as opposed to large ISIs, which skewed the optimization to prefer smaller ISIs and therefore higher firing rates. Including both objectives balanced the difference in bias between them.

The TMS response group related to objectives derived from experimental recordings from the epidural space of the cervical spine of human subjects during single pulses of TMS. The peaks, troughs, and latencies (time-to-peak and time-to-minimum) for each of the corticospinal waves—D-wave (if available), I1-wave, I2-wave, and I3-wave—were measured and used as

**Table 3. List of Optimization Objectives.**

| Objectives | | |
|---|---|---|
| 1. D-wave peak | 18. L2/3 IT ISI | 35. L5 PTN baseline CV |
| 2. D-wave time-to-peak | 19. L2/3 BC firing rate | 36. L5 BC baseline CV |
| 3. D-wave trough | 20. L2/3 BC ISI | 37. L6 IT baseline CV |
| 4. D-wave time-to-trough | 21. L5 PTN firing rate | 38. L6 BC baseline CV |
| 5. I1-wave peak | 22. L5 PTN ISI | 39. L2/3 IT population ISI std. |
| 6. I1-wave time-to-peak | 23. L5 BC firing rate | 40. L2/3 BC population ISI std. |
| 7. I1-wave trough | 24. L5 BC ISI | 41. L5 PTN population ISI std. |
| 8. I1-wave time-to-trough | 25. L6 IT firing rate | 42. L5 BC population ISI std. |
| 9. I2-wave peak | 26. L6 IT ISI | 43. L6 IT population ISI std. |
| 10. I2-wave time-to-peak | 27. L2/3 IT peak/mean ratio | 44. L6 BC population ISI std. |
| 11. I2-wave trough | 28. L2/3 BC peak/mean ratio | 45. L2/3 IT noise weight |
| 12. I2-wave time-to-trough | 29. L5 PTN peak/mean ratio | 46. L2/3 BC noise weight |
| 13. I3-wave peak | 30. L5 BC peak/mean ratio | 47. L5 PTN noise weight |
| 14. I3-wave time-to-peak | 31. L6 IT peak/mean ratio | 48. L5 BC noise weight |
| 15. I3-wave trough | 32. L6 BC peak/mean ratio | 49. L6 IT noise weight |
| 16. I3-wave time-to-trough | 33. L2/3 IT baseline CV | 50. L6 BC noise weight |
| 17. L2/3 IT firing rate | 34. L2/3 BC baseline CV | 51. Amplitude after I3-wave |

objectives. An additional objective minimized the peak of the model output beyond the time-window during which the I3-wave should occur to prevent additional corticospinal waves, which were not present in the recordings.

To reduce population synchrony, the population spiking density for a neuron type was constructed and smoothed with a Gaussian kernel. The ratio between the maximum and the average value and the coefficient of variation of the smoothed population spiking density were used as objectives with target values of one and zero, respectively.

The miscellaneous group included the following objectives. A possible aberrant network behavior resulted in spiking activity of the network being dominated by large firing rates in a few neurons with the remaining neurons being silent. To avoid this, the standard deviation of the mean population ISI within a neuron type was minimized to prevent highly skewed distributions of activity. Another objective acted to identify the minimum noise added to the neurons.

**Optimized parameters.** There were 98 open parameters for optimization. They can be divided into the following categories: Synaptic weights scalars, conduction velocity scalars, afferent delay mean, afferent delay standard deviation, proportion activated, noise amplitude, and noise rate. These categories and their bounds for optimization are summarized in Table 4. The specific names of all parameters are listed in Tables A and B in S1 Appendix.

**Characterizing optimization robustness.** First, individual subject-specific models were optimized. The optimization was repeated five times with different random seeds to increase coverage of the parameter space and avoid local minimum solutions. The final selected models had average corticospinal wave errors of 18.8% and 14.5% for the D+ and D− models, respectively (Fig 4A). The final parameter values for each of the subject-specific models are presented in Figs A-C in S1 Appendix.

Optimizations approached similar total error (Fig D in S1 Appendix). To quantify the similarity of best solutions (i.e., lowest total error) found for each optimization run, the distance among parameters for the best solutions were computed using Euclidean distance, normalized by the maximal possible distance (Fig D in S1 Appendix) with overall distances being 17.4 to 19.6% from each other for D+ and D-, respectively. The relatively low distance (i.e., large similarity) indicated that solutions lie within similar regions of the parameter space.

**Table 4. Categories of optimized parameters.**

| Name | Description | Range |
|---|---|---|
| Synaptic Weight Scalar (N. A.) 38 parameter) | Scalar multiplied to base synaptic weights | [0.1, 10] |
| Conduction Velocity Scalar (N. A.) 24 parameters | Scalar multiplied to conduction velocity | [0.25, 2] |
| Afferent Delay Mean (ms) 6 parameters | Mean conduction delay between afferent and postsynaptic neuron | [0.2, 2] |
| Afferent Delay Stdev. (ms) 6 parameters | Standard deviation of conduction delay between afferent and postsynaptic neuron | [0.1, 1] |
| Proportion Activated (N. A.) 12 parameters | Proportion of population made suprathreshold due to application of TMS | [0, 1] |
| Noise Amplitude (nA) 6 parameters | Amplitude of current to generate spiking activity due to independent noise | [1, 50] |
| Noise Rate (N. A.) 6 parameters | Scalar multiplied with the desired firing rate to determine the mean of the Poisson process used to generate noise | [0, 1] |

When identifying a dominating front, the large number of objectives resulted in every solution being considered dominating. Therefore, the objectives were grouped by category and summed together to reduce the dimensionality of the dominating front to four dimensions. The categories and the corresponding objectives (based on the numbering from Table 3) are the following: Corticospinal wave (1–16), spiking activity (17–26), synchrony (27–38), and well-behaved (39–51). The Pareto front is visualized in Fig E in S1 Appendix. The category error is plotted as a function of total error and showed that corticospinal wave and baseline activity objectives were opposed. Generally, a solution that better matched the experimentally-recorded corticospinal waves had a worse match with the desired baseline activity.

**Identification of a unified model.** Initially, subject-specific models were generated by running the optimization separately for the D+ and D- subjects. However, there were similarities among many of the parameters of the subject-specific D+ and D- models (average relative error of 0.21) that supported the pursuit of a parsimonious model that had identical values for all parameters except for the activation parameters.

The unified model was generated by creating a weighted combination of the parameters of the subject-specific D+ and D- models (Fig 12). The best unified model was selected based on the total error across both subject-specific models as well as the absolute difference of total error between both subject-specific models to identify a unified model that reproduced both response types without favoring one response type over the other. Thus, the unified model could generate responses that were similar to the experimental data for both D+ and D- cases with errors of 18.8% and 24.0% for the D+ and D− responses, respectively (Fig 4). The parameter comparisons, sensitivity analysis, and structural analysis were conducted on the unified model.

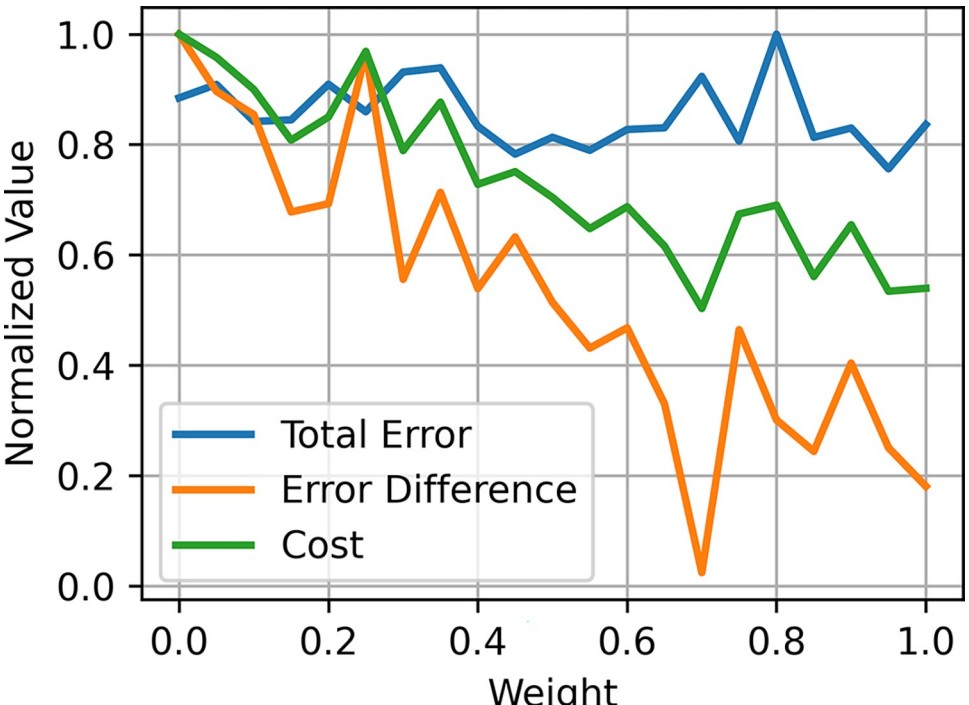

**Fig 12. Unified model search.** The unified model was selected by interpolating between the parameters of the subject-specific D+ and D- models. The cost function for selecting the unified model was the average of total error across both subject-specific models and the absolute difference in errors between both models.

## Sensitivity analysis between model parameters and corticospinal waves

The TVAT analysis investigated the effect of activation parameters and projection strengths on corticospinal wave amplitudes. Activation parameters were varied between 0 and 1, representing no activation to full activation of a population. Synaptic scalars were varied between 0 and 10, representing a lesion of a projection to 10x the strength of the original source model. The 10x upper bound was chosen to compensate for the reduction in model size compared to the original source model. A global sensitivity analysis was chosen over a local sensitivity analysis that may be restricted only to evaluating the robustness of the model rather than be representative of a full characterization of the system. There were 42 total parameters with 21 equally spaced values between 0 and the maximum boundary resulting in 861 unique parameter-pairs with 441 values per pair. The total number of simulations for the sensitivity analysis was 344,400. TVAT simulations were parallelized across 1,000 CPUs with a total execution time of 2,639 compute-hours, using 960 GB of RAM.

The effect size for each parameter on a corticospinal wave amplitude was computed by fitting the surfaces generated by TVAT simulations via polynomial regression and summing the absolute values of the coefficients to represent the effect size. For each pair, the relationships between the two parameters and the amplitudes for each corticospinal wave were approximated using linear regression with elastic net regularization and a third-order polynomial model that included third-order interaction terms. Prior to the linear regression, the corticospinal wave amplitudes were *standardized*, i.e., the mean was subtracted, and the variance normalized to one. Because they were uniformly distributed across a grid, the parameters were *normalized*, i.e., the minimum was subtracted, and the values divided by the parameter boundary range. Regularization is a method of embedded feature selection that determines feature importance during coefficient estimation and prevents overfitting. The optimal regularization parameters were determined using 10-fold cross-validation. The open-source *scikit-learn* Python package was used to conduct the regression and cross-validation [57]. Polynomial regressions of the TVAT surfaces were computed using a single CPU with an execution time of 3.6 compute-hours, using 270 MB of RAM.

The partial effect size of a parameter for a corticospinal wave was represented as the sum of the absolute values of the coefficients of the polynomial models that involved the parameter. The total effect size for a corticospinal wave was calculated as the sum of the effect sizes across all polynomial models, i.e., across all pair-wise interactions, that included the parameter. Poor polynomial fits, indicating that there may be little or no correlation between the parameters and the corticospinal wave amplitude, were excluded from the summation. Only models with a coefficient of determination greater than or equal to 0.5 were included.

## Structural analysis between model circuit and corticospinal wave sensitivity

The cortical column circuit at the neuron population level can be represented as a weighted directed graph with neuron types as nodes and connection between neuron types as edges. Given the effect sizes revealed by the TVAT analysis, classifiers were used to identify any similarities in graph properties that may exist to explain groupings of effect sizes, i.e., preferential versus non-preferential and corticospinal wave preference. The goal was to identify the minimum set of features that would separate preferential vs non-preferential nodes and then identify the corticospinal wave to which a preferential node had the greatest effect.

**Graph metrics.**    Edge weights were characterized using a variety of properties such as conduction delay and the log of the connection probability. Because the relevant output of the network model was generated by the L5 PTNs, graph analysis was conducted using these neurons as a target or reference node. Graph analysis was conducted using the open-source *networkx*

Python package [58]. All simple paths between a starting node and the target node (L5 PTNs) were identified. Simple paths are defined as the sequence of nodes between the start and target that do not include repeat nodes along the path. The total conduction delay from a node to the target was computed as the sum of all conduction delays between nodes along the simple path, including synaptic transmission delays (0.2 ms). The total connection probability was computed as the sum of the logs of all connection probabilities between nodes along the simple path. Averages and standard deviations were also computed for these metrics. The out-degree (divergence), in-degree (convergence), and three centrality measures were calculated as well. Centrality attempts to quantify the importance of a node with different centrality metrics using different criterion. Closeness centrality computes the reciprocal of the average length of the shortest path between a node and all other nodes. Nodes with a higher closeness centrality are "closer" to all other nodes, and their dynamics can propagate more quickly throughout the graph. Harmonic centrality is the average of the inverse of the shortest path between a node and all other nodes, and characterizes sparse networks with greater sensitivity than closeness centrality [59]. Betweenness centrality measures the proportion that a node was included as a part of the shortest path between nodes [60]. Finally, the overall functional effect of the simple path was computed by first determining whether the simple path would have an overall excitatory effect (+1) or inhibitory effect (−1) on the L5 PTNs by multiplying successive functional effects along the simple path. The functional effects of each simple path were then weighted by the log of the path connection probability to compute the weighted average used to represent the overall functional effect of a node to the L5 PTNs. A summary and description of these metrics are in Table 5.

**Training classifiers.**   Two types of classifiers were used based on the number of classes that needed to be identified by the task. Logistic regression was used for binary classification to identify whether an activation had a preferential or non-preferential effect on any corticospinal wave. Support vector classification (SVC) with a radial basis function was used for multiclass classification to identify the corticospinal wave on which a preferential activation had the greatest effect, i.e., the D-wave, I1-wave, I2-wave, or I3-wave [61]. Classification, cross-fold validation, and regularization were conducted using the *scikit-learn* Python package [57].

Each cell type was characterized by a set of features based on the graph metrics described in Table 5 to construct an input matrix. To allow regularization to penalize different types of features in an unbiased manner, each type of feature was *standardized*, i.e., the means were removed, and the variance was normalized to one.

There were only 12 cell types leading to low numbers of examples of each class. This problem was worse for the classification of corticospinal wave preference as there were only 6 cell types with a preferential effect and 4 classes. Therefore, the data was augmented by concatenating noisy versions of the original data. Noise was drawn from a normal distribution with zero mean and a standard deviation of 0.3, which represented 9% of the total variance of the standardized data.

Stratified 10-fold validation with 5 repeats was used to generate training and test sets for validation of the models. Stratified k-fold validation was chosen to allow for a balanced sampling of classes. Classification performance was quantified on the validation sets using accuracy, i.e., the proportion of classifications that were correct. This validation strategy was performed for all the model evaluations described below.

Recursive feature elimination was conducted to identify the most predictive features for each classification problem [61]. During this procedure, an initial random subset of features was chosen, and the classifier was trained and evaluated. Then, classifiers were trained while leaving one feature out. The classifier with the lowest decrease in performance indicated that the removed feature the least predictive and was eliminated from the feature subset. This

**Table 5. Description of graph metrics used to characterize the network.**

| Name | Description |
|---|---|
| Convergence | In-degree of nodes / number of connected presynaptic neuron types. |
| Divergence | Out-degree of nodes / number of connected postsynaptic neuron types. |
| Total Simple Paths | Total number of unique simple paths for a node to L5 PTN. |
| Shortest Path Delay | Conduction delay of shortest path from node to L5 PTN. |
| Average Path Delay | Average path delay of all simple paths from a node to L5 PTN. |
| Weighted Average Path Delay | Weighted average of path delay of all simple paths from a node to L5 PTN using the log of the connection probability of the simple paths as weights. |
| Standard Deviation Path Delay | Standard deviation of path delays of all simple paths from a node to L5 PTN. |
| Weighted Standard Deviation Path Delay | Weighted standard deviation of path delays of all simple paths from a node to L5 PTN using the log of the connection probability of the simple paths as weights. |
| Connection Probability of Shortest Path | Connection probability of shortest path from a node to L5 PTN. |
| Average Connection Probability (Log) | Average of the log of the connection probabilities of all simple paths from a node to L5 PTN. |
| Standard Deviation Connection Probability (Log) | Standard deviation of the log of the connection probabilities of all simple paths from a node to L5 PTN. |
| Functional Effect | Overall excitatory/inhibitory effect of node on L5 PTN. For each simple path the excitatory/inhibitory effect of a node on the next node was represented as a +1 or -1. The effects of successive nodes were multiplied. |
| Weighted Functional Effect | Weighted average of the functional effect using the log of the connection probability of the simple paths as weights. |
| Closeness Centrality [59] | Reciprocal of the average distance of the shortest paths between the node and all other nodes. A larger closeness centrality means that the node is closer to other nodes. $$C_v = \frac{N-1}{\sum_u d(u,v)}$$ where $d(u, v)$ is the shortest path between nodes $u$ and $v$ |
| Harmonic Centrality [59] | Sum of the reciprocal of the shortest path distances between the node and all other nodes. A larger harmonic centrality also indicates that the node is closer to other nodes. $$H_v = \sum_{u|u \neq v} \frac{1}{d(u,v)}$$ where $d(u, v)$ is the shortest path between nodes $u$ and $v$ |
| Betweenness Centrality [60] | Ratio indicating the proportion that a node is included in the shortest path between nodes. $$B_v = \sum_{s \neq v \neq t \in V} \frac{\sigma_{st}(v)}{\sigma_{st}}$$ where $\sigma_{st}$ is the total number of shortest paths from node $s$ to node $t$ and $\sigma_{st}(v)$ is the number of those paths that pass through node $v$. |

process was repeated with the remaining features until the desired number of features remained. In practice, we found two features allowed for good performance with logistic regression while one feature was sufficient for SVC. Recursive feature elimination was repeated 100 times with 5 random features chosen for each iteration. Then, features were ranked by the number of times the feature was the sole remainder after the elimination process and divided by the total number of times the feature was included in a random subset. The regularization weight and the scale factor for the radial basis functions were determined using grid search and cross-validation. The final classifier was trained using Ridge regularization.

Feature selection for logistic regression and SVC was parallelized across 250 CPUs. Their total execution times were 556 and 2,083 compute-hours, respectively, and both used 35 GB of RAM.

## Supporting information

**S1 Appendix. Fig A. Activation and synaptic weight scalar parameters for the unified model**. Fig B. Conduction velocities and afferent activation delays for the unified model. **Fig C. Synaptic noise firing rates and synaptic noise weights for the unified model. Fig D. Characterizations of convergence from optimization. Fig E. Visualization of reduced pareto front. Fig F. Effect sizes for all parameters and all waves. Fig G. Conduction delay histograms resulting from orthodromic and antidromic propagation of action potentials. Table A. List of Optimized Synaptic Weight Parameters. Table B. List of Optimized Delay, Activation, and Noise Parameters.**
(PDF)

## Acknowledgments

The authors thank Dr. Aman Aberra for preliminary work on the I-wave model, and the Duke Compute Cluster team for computational support.

## Author Contributions

**Conceptualization:** Gene J. Yu, Marc A. Sommer, Angel V. Peterchev, Warren M. Grill.

**Data curation:** Federico Ranieri, Vincenzo Di Lazzaro.

**Formal analysis:** Gene J. Yu.

**Funding acquisition:** Marc A. Sommer, Angel V. Peterchev, Warren M. Grill.

**Investigation:** Gene J. Yu.

**Methodology:** Gene J. Yu, Angel V. Peterchev, Warren M. Grill.

**Project administration:** Angel V. Peterchev, Warren M. Grill.

**Resources:** Angel V. Peterchev, Warren M. Grill.

**Software:** Gene J. Yu.

**Supervision:** Angel V. Peterchev, Warren M. Grill.

**Validation:** Gene J. Yu.

**Visualization:** Gene J. Yu.

**Writing – original draft:** Gene J. Yu.

**Writing – review & editing:** Gene J. Yu, Federico Ranieri, Vincenzo Di Lazzaro, Marc A. Sommer, Angel V. Peterchev, Warren M. Grill.

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
