## [Decision Letter · Decision Letter 0]

18 Jun 2024

Dear Dr Yu,

Thank you very much for submitting your manuscript "Circuits and mechanisms for TMS-induced corticospinal waves: Connecting sensitivity analysis to the network graph" for consideration at PLOS Computational Biology.

As with all papers reviewed by the journal, your manuscript was reviewed by members of the editorial board and by several independent reviewers. In light of the reviews (below this email), we would like to invite the resubmission of a significantly-revised version that takes into account the reviewers' comments.

We cannot make any decision about publication until we have seen the revised manuscript and your response to the reviewers' comments. Your revised manuscript is also likely to be sent to reviewers for further evaluation.

Sincerely,

Hermann Cuntz

Academic Editor

PLOS Computational Biology

Daniele Marinazzo

Section Editor

PLOS Computational Biology

Reviewer's Responses to Questions

**Comments to the Authors:**

Reviewer #1: I am not an expert in the computational methods used in this paper and therefore my comments are directed at the neurophysiological details and conclusions.

Physiologically the contribution of the paper is moderate since, as the authors acknowledge, some of the outcomes (e.g. the role of GABAa inhibitory neurons in the I1-wave) do not match known physiology. Its main impact, I think, will be on how this type of relatively unconstrained model with so many adjustable parameters can nevertheless generate some very reasonable conclusions. It is also the first paper to model explicitly the actual outcome of experiments conducted in humans. In this respect it is very impressive. On the negative side, the results are limited to the response of the model to a single stimulus of a single intensity. Behaviour of I-waves at different intensities and in response to double pulse experiments is already known in the physiological literature, so in theory these could be modelled in future papers (see limitations section of the present paper).

I have the following comments for the authors.

1) I was slightly surprised to see that despite the fact that recordings were taken from sites several cm apart (C1-C2 versus C3-C5), that the latencies of the I-waves in Fig 1B were so similar. Surely the C3-C5 site should be slightly delayed with respect to the C1-C2 site?

2) I have presumed that the TMS pulse activates neurons at presynaptic boutons and not cell bodies (hence excludes axon conduction times), apart from TMS activation of PTN neurons which I suppose activates the axon hillock. Is this correct?

3) In the Esser model, L6 inputs to L5 are relatively weak compared with those from L2/L3, which I think is correct physiologically. As far as I can see this is not the case in the present model. Perhaps as a result of this, input from L6 plays an important role in production of I2 and I3 waves. Previous physiological models have generally suggested that input from L2/3 would play a more important role. Could the authors comment?

4) Fig A suggests that L6BC show greater activation in the D+ model than the D- model. Given that these synaptic inputs probably arrive too late to influence D-waves, why is this the case? I think its necessary not only for a computational model to work but also that we understand why it works.

5) Similarly, why does the L2/3 BC AFF – L2/3 BC play a role in I1-wave generation (Fig 6) given that its contribution should arrive too late to influence the main monosynaptic input from L5 AFFs?

6) For interest, a paper by Edgley et al (DOI: 10.1093/brain/120.5.839) notes that motor cortical PTN neurons may fire multiple times in response to a TMS pulse, which seems to be in contrast to the model predictions here.

Reviewer #2: In the manuscript entitled: “Circuits and mechanisms for TMS-induced corticospinal waves: Connecting sensitivity analysis to the network graph,” the authors presented a detailed study of the mechanisms underlying D- and I-wave generation and motor cortical stimulation via TMS. The methodological approach to investigate the mechanisms is built on a well founded model structure and includes optimization against real patient data, sensitivity analysis, and structural analysis. Studying the model behavior across a wide range of well defined parameters and constraints was a challenging task which the authors approached well. The study was also very well visualized and presented in the manuscript. In order to further improve the paper, I have the following suggestions:

Major comments:

-------------------

#133: The Results section contains a lot of methods. I strongly suggest to leave out methods details in results, or refer to the methods supplying minimal detail in results. e.g how is the tms defined, input-output approach (just one example: #192-199)

#174: Whole section on unified models needs to be clarified. It is unclear where a weighted combination of parameters is taken, or parameters are identical, or the best individual parameters from one model are taken. Whenever the ‘models’ are referred to it is not exactly clear what is meant. Also ‘unified D- model’ and ‘unified D+ model’ are confusing #186, #187. Reconsider notation for each model.

#201: a sensitivity analysis of such a complex problem is usually conducted in close vicinity around the working point. Analyzing the sensitivity in the entire space is potentially not telling because it shows the global behavior including a lot of cases far from reality. These unrealistic cases have a big influence on the final sensitivities. When decreasing the range around the working point, the polynomial fits will get way better and the sensitivity coefficients will be more accurate. If possible, it would be very interesting to analyze the sensitivities in a smaller range and to compare them with the ones presented in the current manuscript.

Fig. 8: From these observations, it seems that the results did not converge yet with the selected time step of 0.025 ms (see number of spikes, ISI, etc.). From the convergence results you show, a step size of 0.01 ms would be more appropriate to ensure stable results while not being completely infeasible to realize I suppose. It leaves a discomforting feeling when interpreting the results. Please show that the final results, i.e. sensitivities etc., are not affected by the chosen stepsize.

#327-328, #364-368: Please relate these important findings to the different hypotheses discussed in Ziemann (2020) to shed more light onto the potential mechanisms of I wave generation:

Ziemann, U. (2020). I-waves in motor cortex revisited. Experimental brain research, 238(7), 1601-1610.

#435: The Limitations Section needs to be extended:

- mention that the model is not capable of describing effects of directional sensitivity because E-field coupling is simplified (effects TMS coil orientation, PA vs AP etc.)

the D+ and D- experimental results come from different subjects, and it's not discussed how experimentally one can tune D-wave activation. (from recruitment/dose dependence and TMS coil orientation)

- there is no time dynamics to the stimulation (biphasic vs monophasic pulses), please discuss how your definition of current pulses and TMS activation could be generalized to specific waveforms (or not).

- regarding dosage dependence (which you already mention) you could add that this kind of sensitivity analysis you did has to be actually done for every stimulation intensity because the mechanisms will change across the IO curve. If done, you could show a shift of mechanisms across intensities.

#481: Conclusion Section is very vague, and non-specific to the detailed conclusions drawn throughout the results section. Please extend and be more specific.

#775: The Training Classifiers section is written quite abstractly, making it difficult to directly relate the machine learning methods to the details of your model. For example:

#784: Please define “class” and “samples” related to your model.

#789: Please define your quantity of interest how you define true positive and true negative you validate against.

#806 Table 5: The definitions of the graph metrics are quite difficult to understand. Some would benefit from different wording, or potentially using mathematical notation

Minor comments:

-------------------

Clarify the difference between the machine learning model (sometimes called a neural network in some applications) and the point neuron network (aka Esser), to be sure that the distinction is clear between the machine learning methods deployed and the computational model developed.

#88 - 94: - clarify the definitions of each hypothesis

#100 - 104: - rephrase talking about constraints to recordings. Constraint to recordings is not necessarily an advantage rather a different approach that works forward from first principles to DI-waves rather than fitting/exploring parameters strictly constrained to experiment.

#165 Fig 4 caption, ‘unified D– model’ is confusing.

#179 - 180: Stick to one notation for (D+ and D-) or (D-wave and non-D-wave) model

#445: Please motivate the choice for the factor of 1800. I suppose it is something like the average number of segments in a compartment model but it comes a bit out of the blue here.

#516: ‘matched the range of microcolumns per macrocolumn’ may be more clear with ‘range’ replaced by ‘ratio.’

#540 - 542: Could be rephrased for clarity. Based on the specified activated proportion from stimulus for the chosen afferent or neuron type, a corresponding proportion of that given population or afferent type was randomly chosen to be presented with a stimulus.

#543: Please provide information about the current pulse, i.e. pulse duration and amplitude.

#548 - 549: Reading that connectivity rules for each microcolumn are identical, it is now unclear to me whether connections are only formed within microcolumns or also across them.

#553: I suggest to reformulate this sentence (measurements -> observations, simulations, or stimulations)

#565: Is 20 seconds the actual time it takes to run the model once given some PC hardware? If so please provide details about the PC hardware. Or is it the simulated time in the model to study the time steps in terms of diverging solutions? In #560 you write that the total simulated time is 3000 ms in the model. Please be more clear here and try to reformulate “simulation time” in the real world to run the model and “simulated time” inside the model. I’m confused. If not already done please also report the actual simulation time on the used PC hardware.

#569: Please describe shortly how the van Rossum spike distance is defined.

#577: in Fig. 8 it can be seen from “Number of Spikes that the knee is below 0.03 ms. Why was 0.03 chosen as the reference (see major comment before).

Fig 8: “Normalized” and “Difference” curves need more explanation in the main text. Please also indicate with a dashed (colored) line also the final step size you selected (0.025). Please label the y-axis and units of the Coefficient of Variation (ISI) mean plot (middle left)

#608: Please report the specific coil type

#609: I suggest rephrasing “optimal scalp position” here because no one really knows where the optimal location really is.

#616-617: Is it not a concern that the D+ and D- cases are from different subjects? (-- Aaron to Konstantin)

#628: It is not clear to me what ‘best solution found by itself’ means. What is ‘it?’

Fig 9: ‘Generation’ on the x-axis is not mentioned in the main text, and ‘iteration’ is used in the figure caption. Would be best to pick one term for consistency.

#655: ‘optimization parameters’ -> ‘optimization hyperparameter’ could clarify these are independent of the optimized parameter space.

#669: Clarify the calculated difference here, unclear what this sentence means.

#673: Please describe the constraint category “well-behaved” in a bit more detail.

#693 - 694: It may be prudent to mention somewhere that by only optimizing against recordings with 3 I-waves, the model is not trained to exhibit possible further bursts, and assumes only these 3 I-waves can exist.

#728: It sounds like you used a generalized polynomial chaos expansion of order 3 and extracted the Sobol indices from it (#742). Is this correct?

#767 - 768: What are the divergence, convergence, and centrality measures calculated? Is this obvious? Ah ok this is mentioned in Table 5

Table 5: Some of these written descriptions are quite confusing. Especially Harmonic Centrality, which I needed to try and work out in summation notation to understand (maybe). It is possible that some of these descriptions would be better suited for mathematical/symbolic notation than as written descriptions.

Reviewer #3: The authors developed a point-neuron network model of the human motor cortical macrocolumn to simulate realistic D-waves and I-waves in response to single-pulse TMS. This model, based on a previous cortical model by Esser et al, allowed D-waves to be included or excluded by adjusting the activation of L5 PTNs. The 98 model parameters (related to synaptic weights, conduction delays, afferent delays, proportion of neuronal activations and noise) were optimised to achieve good performance for several constraints, including (asynchronous) baseline activity as well as TMS responses in the form of corticospinal waves (constrained by human single-pulse TMS data).

First, the authors performed a sensitivity analysis to identify key afferents and neuron types within the motor cortex that generate corticospinal waves when activated. The sensitivity analysis showed that activation of the L5 PTNs mainly influenced D-waves, while afferents to the L5 PTNs significantly influenced the I1-wave. Direct activation of afferents was crucial for the generation of all I-waves, contradicting the idea that I-waves are generated by repetitive firing of individual neurons. Interestingly, the I3 wave was more sensitive to afferent input, the I1 wave was more sensitive to intracortical synaptic parameters, whereas the D wave showed no sensitivity to synaptic parameters. The waves were equally sensitive to excitatory and inhibitory neurons.

Secondly, the authors addressed the issue of degeneracy revealed by the sensitivity analysis (degeneracy in the sense of multiple different mechanisms contributing to cortico-spinal waves). To this end, the authors performed a structural analysis based on graph theory and machine learning (logistic regression with lasso regularisation, recursive feature elimination with support vector classification). The structural analysis focused on the wiring diagram and conduction latencies and helped to find common network features that contribute to cortico-spinal wave generation. Neuron types with high connectivity to L5 PTNs were found to contribute significantly to I-waves. High connection probabilities to L5 PTNs and the conduction delay of the shortest path to L5 PTNs were critical factors in determining the latency of an I-wave. Interestingly, inhibitory interneurons influenced several I-waves.

This is a solid and carefully conducted and interpreted computational study, using advanced methods to optimise parameters and estimate causal relationships between the many model features and the successful simulation of experimental results.

Major issues:

- Were the successful parameters (e.g. conduction velocities, afferent delays, conduction delays) within a plausible biological range? See also next question.

- Line 542: „Direct activation of neurons resulted in an injection of a short suprathreshold current to elicit an action potential that was propagated orthodromically to all postsynaptically connected neurons using all relevant conduction delays. Direct activation of the terminals of afferents resulted in the activation of all connected synapses with the appropriate conduction delays.“ Current theory and models of TMS (see e.g. Siebner et al. Clin Neurophysiol 2022) assume that spikes are triggered by TMS in the axon terminals (not in the somata). It seems that in your simulations of direct activation of neurons you assumed direct somatic activation and subsequent orthodromic spike propagation along the axon (simulated implicitly in the conduction delays). Can you comment on this? How would your results change if you assumed that TMS induces axonal spikes in the axonal tips (instead of somatic spikes that then propagate orthodromically), and if you shortened the conduction delay by omitting the orthodromic spike propagation? If this has a bearing on the interpretation of the results, you should include a short paragraph mentioning this.

- The authors write (l. 725): „Generally, a solution that better matched the experimentally-recorded corticospinal waves had a worse match with the desired baseline activity.“ Can the authors briefly discuss in the paper how to improve this trade-off in the performance of the model or – if I have missed it - can they give me the line numbers where they have already done this?

- The authors used a relatively old M1 point-neuron model (Esser et al. 2005). They had good reasons for doing so (ability to simulate I-waves), but as an outlook they could mention newer, much more realistic models that could be used in the future for TMS modelling - e.g. Bill Lytton's model (https://pubmed.ncbi.nlm.nih.gov/37300831/).

- The authors did not explicitly model the electric fields generated by TMS (see e.g. Mantell et al. Neuroimage 2023), but used direct activation of neurons and afferents as a proxy for TMS. A suggestion: in the discussion, the authors could briefly mention a possible future extension or combination of their network modelling with recently emerging multi-scale TMS modelling approaches (Aberra et al. J neural Eng 2018, Shirinpour et al. Brain Stimul. 2021, Weise et al. Imaging Neuroscience 2023), e.g. applied to more realistic M1 models (see above). It would be potentially very interesting to combine such multi-scale TMS electric field modelling with anatomically highly detailed M1 models and with the authors' powerful sensitivity analysis and machine learning and graph theory based structural analysis.

- Could the authors comment on the mechanistic explanations for the effects of inhibitory connections in the model - in terms of feed-forward vs. feed-back inhibition effects? Is it possible in their model to disentangle the contributions of FF and FB inhibition to corticospinal waves? Would the addition of SOM, VIP inhibitory motifs change the dynamics of the model and possibly its conclusions (e.g. Lytton's M1 model mentioned above includes somatostatin-expressing (SOM) interneurons)? TMS-induced excitation of excitatory neurons depends on the functional state of the stimulated network, which depends on the overall inhibition mediated by the plethora of inhibitory interneurons.

Minor issues:

- Line 502, 503 and line 565, 566: „The spiking activities of the afferents were generated by a Poisson process with a mean firing rate of 0.25 Hz“ „the models received a random Poisson input with a mean firing rate of 1000 Hz“

A 1000 Hz Poisson input does not seem to be biologically realistic. Can you explain this and the discrepancy between the two sentences?

- The authors write: „The predictions from the model are limited to the single pulse response and are not readily extendable to paired pulse or repetitive pulse paradigms. This is partly due to GABABR parameters being underconstrained.“ Another important limitation is the lack of short-term synaptic plasticity mechanisms in the simple network model used. And see also https://doi.org/10.1016/j.neuron.2024.05.009 for cell-type-specific electric field entrainment, which may also affect repetitive stimulation paradigms.

**Have the authors made all data and (if applicable) computational code underlying the findings in their manuscript fully available?**

Reviewer #1: Yes

Reviewer #2: Yes

Reviewer #3: Yes

PLOS authors have the option to publish the peer review history of their article (what does this mean?). If published, this will include your full peer review and any attached files.

Reviewer #1: No

Reviewer #2: **Yes: **Dr. Konstantin Weise

Reviewer #3: No
---

## [Decision Letter · Decision Letter 1]

14 Nov 2024

Dear Dr Yu,

We are pleased to inform you that your manuscript 'Circuits and mechanisms for TMS-induced corticospinal waves: Connecting sensitivity analysis to the network graph' has been provisionally accepted for publication in PLOS Computational Biology.

Best regards,

Hermann Cuntz

Academic Editor

PLOS Computational Biology

Daniele Marinazzo

Section Editor

PLOS Computational Biology

Feilim Mac Gabhann

Editor-in-Chief

PLOS Computational Biology

Jason Papin

Editor-in-Chief

PLOS Computational Biology

Reviewer's Responses to Questions

**Comments to the Authors:**

Reviewer #1: Thanks for the detailed responses to my queries. They are most helpful and I'm happy with the revisions that include them.

Reviewer #2: The authors carefully addressed all of my comments. I appreciate that they repeated the analysis with a descreased step size of 0.01!

Reviewer #3: The authors have addressed my concerns.

**Have the authors made all data and (if applicable) computational code underlying the findings in their manuscript fully available?**

Reviewer #1: Yes

Reviewer #2: **No: **I recommend to make the optimization routine and computational code available.

Reviewer #3: Yes

PLOS authors have the option to publish the peer review history of their article (what does this mean?). If published, this will include your full peer review and any attached files.

Reviewer #1: No

Reviewer #2: No

Reviewer #3: No

---

## [Editor Report · Acceptance letter]

26 Nov 2024

PCOMPBIOL-D-24-00373R1 

Circuits and mechanisms for TMS-induced corticospinal waves: Connecting sensitivity analysis to the network graph

Dear Dr Yu,

I am pleased to inform you that your manuscript has been formally accepted for publication in PLOS Computational Biology. Your manuscript is now with our production department and you will be notified of the publication date in due course.

With kind regards,

Anita Estes
